# Invadopodia enable cooperative invasion and metastasis of breast cancer cells

Louisiane Perrin [1,3], Elizaveta Belova[1], Battuya Bayarmagnai[1], Erkan Tüzel[1] & Bojana Gligorijevic [1,2✉]

Invasive and non-invasive cancer cells can invade together during cooperative invasion. However, the events leading to it, role of the epithelial-mesenchymal transition and the consequences this may have on metastasis are unknown. In this study, we demonstrate that the isogenic 4T1 and 67NR breast cancer cells sort from each other in 3D spheroids, followed by cooperative invasion. By time-lapse microscopy, we show that the invasive 4T1 cells move more persistently compared to non-invasive 67NR, sorting and accumulating at the spheroid-matrix interface, a process dependent on cell-matrix adhesions and independent from E-cadherin cell-cell adhesions. Elimination of invadopodia in 4T1 cells blocks invasion, demonstrating that invadopodia requirement is limited to leader cells. Importantly, we demonstrate that cells with and without invadopodia can also engage in cooperative metastasis in preclinical mouse models. Altogether, our results suggest that a small number of cells with invadopodia can drive the metastasis of heterogeneous cell clusters.

[1] Bioengineering Department, Temple University, Philadelphia, PA, USA. [2] Cancer Signaling and Epigenetics Program, Fox Chase Cancer Center, Philadelphia, PA, USA. [3] Present address: Institut Curie, UMR144, Paris, France. ✉email: bojana.gligorijevic@temple.edu

More than 90% of cancer patients die due to complications resulting from metastasis, i.e. the process of dissemination, re-seeding and growth of cancer cells in secondary organs[1,2]. In breast cancer, recent work demonstrated that metastases mostly arise from polyclonal seedings[3,4]. These polyclonal metastases develop from the collective dissemination of cell clusters, as opposed to the successive accumulation of multiple single clones. Together with the growing literature on phenotypic heterogeneity within primary tumors[5], these observations suggest that cooperativity between clones of cancer cells may facilitate metastasis.

In breast cancer, the metastatic cascade is initiated when cancer cells acquire invasive properties, which includes the integration of motility and the ability to degrade the extracellular matrix (ECM)[2]. Both invasion and motility are commonly associated with the activation of the epithelial-mesenchymal transition (EMT) program[6]. During EMT, epithelial cells gradually lose cell-cell contacts, progressively strengthen their adhesions to the ECM and increase their contractility, becoming motile. Concomitantly to EMT, cancer cells can also acquire the capacity to locally degrade the ECM using invadopodia[7–10]. Invadopodia are membrane protrusions enriched in matrix metalloproteinases (MMPs) that confer cancer cells with high proteolytic activity[11,12] and importantly, elevated metastatic potential[13,14]. Since the EMT program is not a binary, nor a unidirectional switch, and since multiple EMT routes exist, distinct EMT trajectories may result in cancer clones with different levels of invasiveness[15].

Recent 3D in vitro work on breast cancer showed that invasive collective strands are composed of cancer cells that differ in multiple invasive traits[3,16–20]. For example, compared to follower cells, leader cells show increased contractility[19], cell-ECM adhesion[3,16], ECM remodeling[20] and ECM degradation capacities[18,19]. As a result, leader cells can enable the invasion of otherwise non-invasive follower cells, a phenomenon termed cooperative invasion[21]. Our recent work showed that leader cells largely reside in the G1 phase of the cell cycle, the phase of the cell cycle during which invadopodia-mediated ECM degradation is the highest[22]. These data suggest that during collective invasion, leader cells may preferentially assemble invadopodia. Although breakdown of the ECM is clearly required for collective invasion, the role of invadopodia-mediated ECM degradation in leader vs. follower cells is unclear. To this date, none of the studies detailed the spatial reorganization that may precede the cooperative invasion. Furthermore, as all the previous studies investigated cooperative invasion in 3D cultures, the cooperation during dissemination and metastasis were not yet explored.

In this study, we aim to understand how cancer clones with differential invasive skills may cooperate during invasion and metastasis. We show that invasive clones can sort from and lead non-invasive clones into cooperative invasion and metastasis. Our study suggests that cooperativity between cancer cells may be an efficient mechanism for collective metastasis.

## Results

**4T1 cells, but not 67NR cells, invade into collagen I.** To investigate how cancer clones with differential invasive skills cooperate during metastasis we used the isogenic pair of breast cancer cell lines 4T1 and 67NR, syngeneic to Balb/C mice[23,24]. Upon orthotopic implantation, both cell lines grow primary tumors, but only 4T1 cells metastasize[25]. We assessed the invasive capacities of 4T1 and 67NR cells in the spheroid invasion assay in the high-density collagen I, which requires MMP-driven degradation of the matrix[22]. After two days, we found that the 4T1 cells exhibited robust invasion into the collagen I matrix, while 67NR cells did not invade (Fig. 1a, b). Treatment with the pan-MMP

inhibitor GM6001 effectively blocked invasion of 4T1 cells (Fig. 1a, b). In addition, immunofluorescence labeling of MMP-mediated collagen I cleavage sites (Col ¾) showed that the invasion of 4T1 cells into the matrix was MMP-dependent (Fig. 1c). Treatment with mitomycin C, which impairs cell division[26], confirmed that the invasion of 4T1 cells was not due to cell proliferation (Figs. 1a, b and S1). 4T1 cells are known to invade as collective strands, with presence of E-cadherin at cell-cell junctions[27,28]. We confirmed that 4T1 cells expressed E-cadherin, while 67NR cells expressed N-cadherin (Fig. S2a, b)[27–29]. Both 4T1 and 67NR cell lines expressed vimentin (Fig. S2a). Interestingly, on the EMT axis (phenotypic continuum from epithelial to mesenchymal), this classifies the invasive 4T1 cells as epithelial/mesenchymal and the non-invasive 67NR cells as mesenchymal. Further, in the 4T1 spheroids, we found E-cadherin to be enriched at all cell-cell junctions, namely between leader and follower cells as well as between follower and follower cells. This verified that the integrity of E-cadherin-mediated cell-cell junctions was maintained during invasion (Fig. S2c, d)[30]. Overall, these results demonstrated that 4T1 cells perform MMP-dependent collective invasion, while the 67NR cells do not invade into dense collagen I matrix.

Invadopodia are membrane protrusions enriched in actin, actin-binding proteins, such as cortactin and Tks5, and MMPs[11,12]. As invadopodia function results in local ECM degradation, we hypothesized that invadopodia play a role in the invasion of 4T1 cells. We also reasoned that the observed difference in the invasion capacities of 4T1 vs. 67NR cells might be explained, at least in part, by a disparity in their invadopodia function. To test this, we first analyzed the expression level of key invadopodia components cortactin and Tks5[11]. Both cell lines expressed similar levels of cortactin and Tks5 (Fig. 1d). We next measured invadopodia function by culturing cells on top of fluorescently labeled gelatin, which allows visualization of degradation as holes in the matrix[31]. We found that 4T1 cells, but not 67NR cells, were able to degrade the gelatin layer (Fig. 1e, f). Puncta of co-localized Tks5, F-actin and degraded gelatin, indicative of functional and mature invadopodia, were present in 4T1 cells (Fig. 1g). This suggests that the observed degradation holes were generated by invadopodia. Puncta of co-localized Tks5 and F-actin, indicative of invadopodia precursors, were present in both 4T1 and 67NR cells, at similar levels (Fig. S3a, b). Altogether, these results suggests that invadopodia precursor fail to mature in 67NR cells. To examine whether invadopodia also play a role in the invasion of 4T1 cells in spheroids, we labeled spheroids for F-actin, Tks5 and MMP-mediated collagen I cleavage sites. We identified functional invadopodia in leader cells, demonstrated by co-localization of F-actin, Tks5 and MMP-mediated collagen I cleavage sites (Fig. 1h). These observations established a link between invadopodia and collective invasion of 4T1 cells and suggest that leader cells assemble invadopodia.

Since the invasion phenotype consists of invadopodia-mediated ECM degradation and cell migration[32], we performed a scratch assay to investigate whether 67NR cells can migrate as efficiently as 4T1 cells. We tracked individual cells (Fig. 1i, j and Movie S1) and found that while the instantaneous speed of 67NR cells is higher than that of 4T1 cells (Fig. 1k), 4T1 cells are significantly more persistent than 67NR cells (Fig. 1l).

In summary, we showed that the 4T1/67NR pair is a suitable tool to investigate how cancer clones with differential invasive capacities cooperate during invasion.

**Persistence drives cell sorting between 4T1 and 67NR cells.** We next set out to investigate dynamics of spatial organization and invasion in the spheroids where 4T1 and 67NR are mixed together. We generated 4T1-mScarlet and 67NR-GFP cell lines and mixed

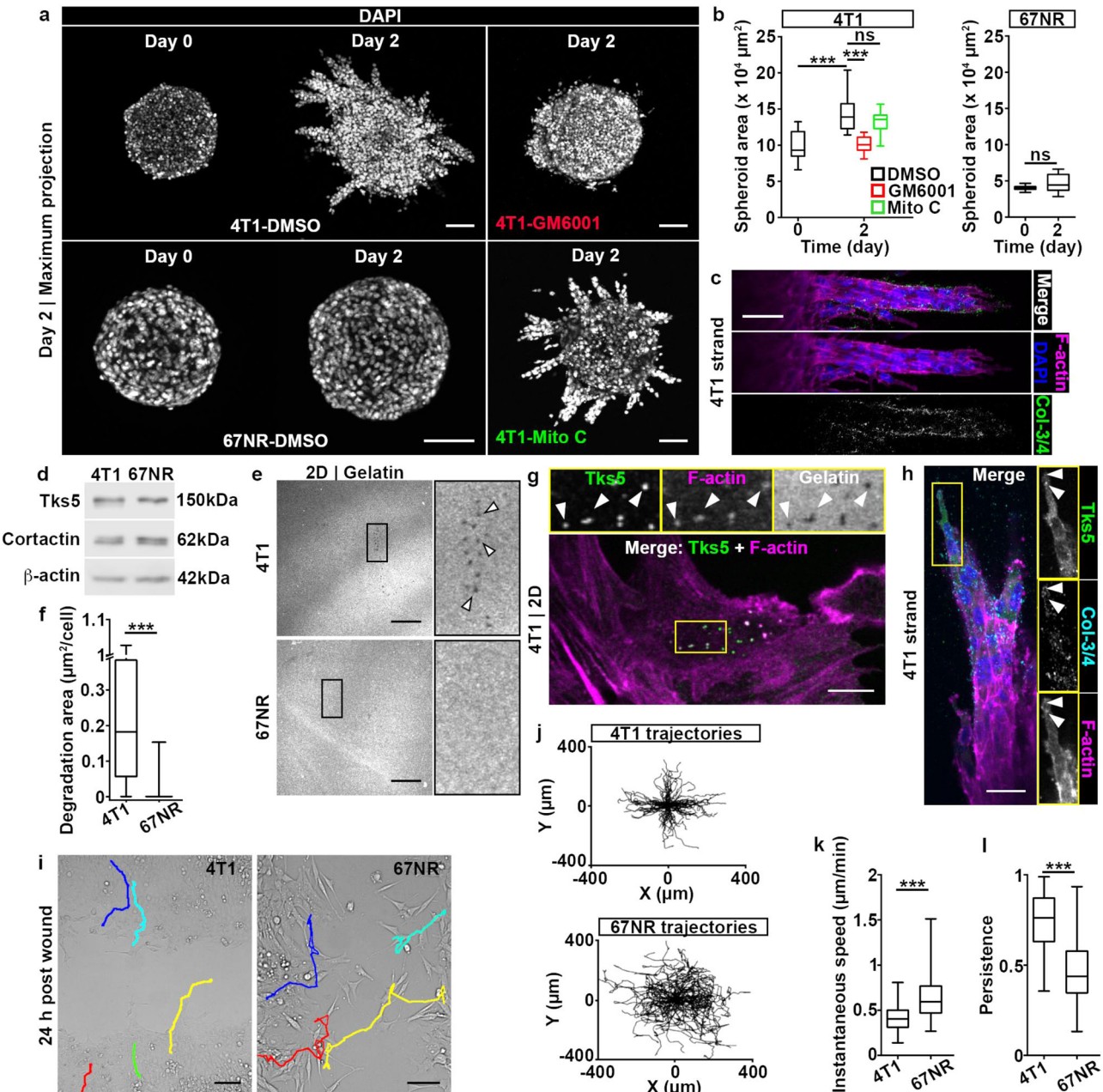

**Fig. 1 4T1 cells, but not 67NR cells, exhibit MMP-dependent spheroid invasion. a** Spheroids of 4T1 and 67NR cells at day 0 and day 2 post-embedding in a 3D collagen I matrix. Nuclei were stained with DAPI. Spheroids were treated from day 0 with a pan-MMP inhibitor (GM6001, top right panel), a cell cycle inhibitor (mitomycin C, Mito C, bottom right panel) or DMSO control (left panels). Scale bars: 100 μm. **b** Spheroid area as a function of time for 4T1 and 67NR cells from (**a**). $P = 5.81 \times 10^{-4}$ and $1.23 \times 10^{-4}$, by the $t$-test. **c** Invading strand of a 4T1 spheroid, day 2 post-embedding, immunolabeled for MMP-cleaved collagen I (Col-3/4, green) and stained for F-actin (phalloidin, magenta) and nuclei (DAPI, blue). Scale bar: 50 μm. **d** Western blot of Tks5 and cortactin expression in 4T1 and 67NR cells. β-actin is used as a loading control. **e** Gelatin degradation for 4T1 (top panel) and 67NR (bottom panel) cells 18 h post-plating. The insets show a 4X zoom-in of the boxed area and arrowheads indicate representative degradation holes. Scale bars: 20 μm. See Fig. S3a for F-actin. **f** Degradation area for 4T1 and 67NR cells from (**e**). $P < 4.40 \times 10^{-16}$, by the Wilcoxon rank sum test. **g** 4T1 cells cultured on fluorescent gelatin (gray), labeled for Tks5 (green) and F-actin (phalloidin, magenta). The insets show a 2X zoom-in of the boxed area and arrowheads indicate representative functional invadopodia. Scale bar: 10 μm. **h** 4T1 strand on day 2 days post-embedding, labeled for Tks5 (green), cleaved collagen (cyan), F-actin (magenta) and nuclei (blue). The insets show a 1.25X zoom-in of the boxed area. Scale bar: 30 μm. **i** 4T1 (left) and 67NR (right) monolayers 24 h post-wounding. Representative cell trajectories are shown. See Movie S1. Scale bars: 100 μm. **j** Trajectories of 4T1 (top) and 67NR (bottom) cells from the wound assay in (**i**), shown as wind-rose plots shifted to a common origin. **k** Instantaneous speed of 4T1 and 67NR cells from (**j**). $P = 1.17 \times 10^{-13}$, by the Wilcoxon rank sum test. **l** Persistence (net displacement/path length) of 4T1 and 67NR cells from (**j**). $P < 2.20 \times 10^{-16}$, by the Wilcoxon rank sum test. Uncropped western blots are available in Supplementary Fig. S13.

them at a 1:50 ratio, to account for the higher proliferation rate of 4T1 over the course of the spheroid invasion assay (Fig. S4a, b). We next embedded the mixed spheroids in collagen I, and performed daily longitudinal imaging (Figs. 2a and S4c)[33]. We noticed that

67NR and 4T1 cells sorted from each other starting on day 3. Individual optical slices (Movie S2) and the analysis of cell coordinates clearly demonstrated the enrichment of 4T1 cells at the edge of spheroids (Figs. 2b and S4d). To quantify cell sorting, we

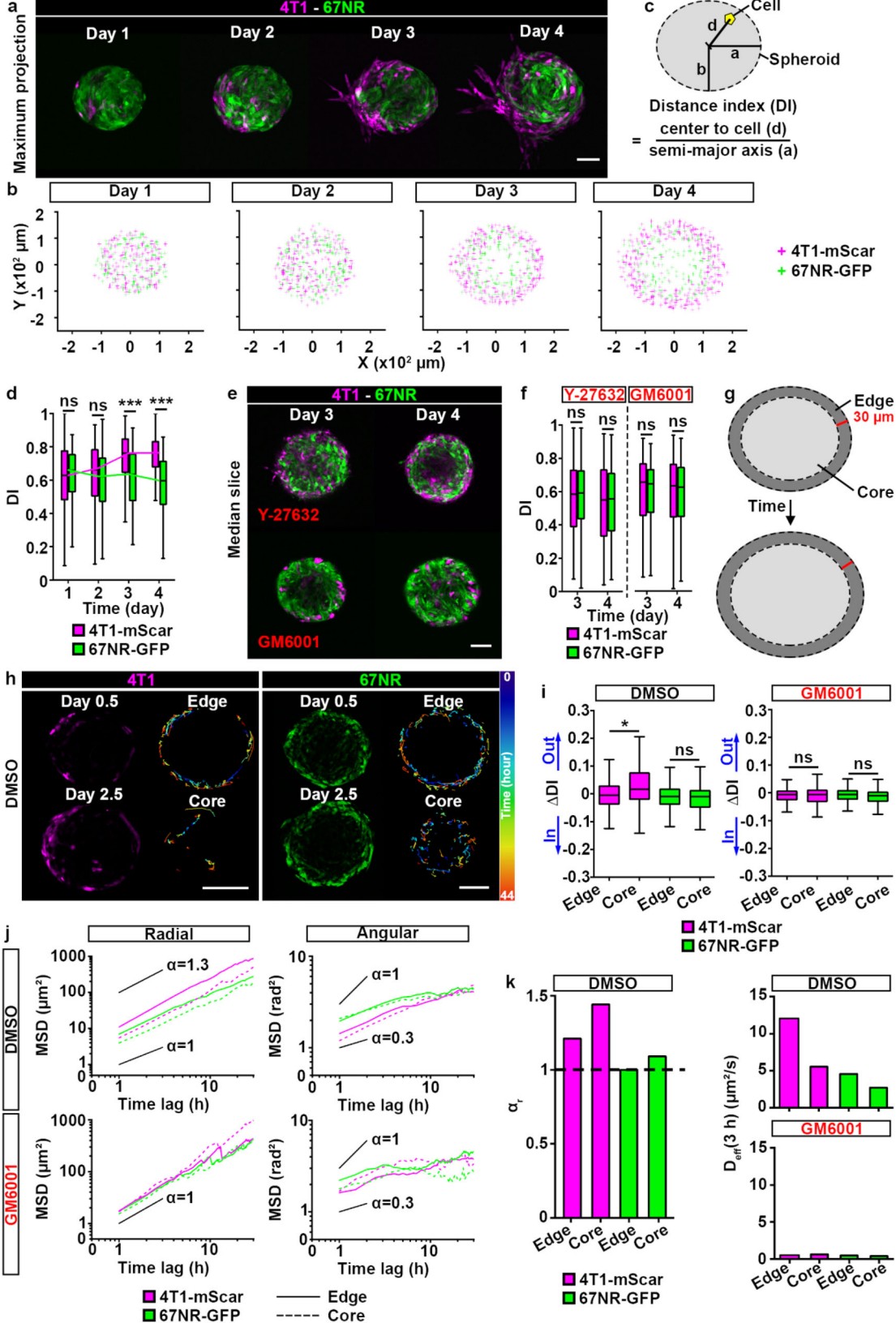

calculated the relative distance of each cell to the spheroid center, a metric we named *Distance Index* (DI), such that a value of 0 marks a cell at the spheroid center and a value of 1 corresponds to a cell at the spheroid-collagen I interface (Fig. 2c). At day 3 post-embedding, we found that the DI of 4T1 cells increased over time and was significantly higher than the DI of 67NR cells (Fig. 2d). This trend was also present in individual spheroids (Fig. S4e). These data revealed that, over the course of 3 days, cells reorganized from a random distribution to spheroids with 4T1 cells populating the interface and 67NR cells located in the spheroid core. On days 4-6, cell sorting of 4T1 and 67NR cells was followed by the invasion (Fig. S4f).

**Fig. 2 Prior to invasion, 4T1 cells sort from 67NR cells via differential directed motility. a** Mixed spheroid, at a 1:50 ratio of 4T1-mScarlet (magenta) to 67NR-GFP (green) cells, embedded in 3D collagen I and imaged daily. Day 1 indicates day 1 post-embedding. See Movie S2. Scale bar: 100 μm. **b** Coordinates of 4T1-mScarlet (4T1-mScar, magenta) and 67NR-GFP (green) cells from all spheroids presented in (**a**) and Fig. S3c. **c** Schematic representation of the distance index (DI); a and b represent the semi-major/minor axes of the spheroid and d represents distance between the spheroid center and a cell. DI is d/a, the relative distance of each cell to the spheroid center, see Materials and Methods. **d** DI for 4T1-mScarlet (magenta) and 67NR-GFP (green boxes) cells from spheroids in (**a**, **b**). $P = 1.32 \times 10^{-14}$ and $<2.20 \times 10^{-16}$, by the Wilcoxon rank sum test. **e** Mixed spheroids at day 3 and 4 post-embedding. Spheroids were treated from day 0 with an inhibitor of cell contractility (ROCK inhibitor, Y-27632, top panels) or a pan-MMP inhibitor (GM6001, bottom panels). Scale bar: 100 μm. **f** DI for 4T1-mScarlet (magenta) and 67NR-GFP (green) cells from spheroids in (**e**). **g** Schematic of the edge (dark gray) and core (light gray) compartments in a spheroid. **h** Snapshots of a mixed spheroid taken at the beginning (day 0.5, 10 h post-embedding) and end of time-lapse recording (day 2.5, 54 h post-embedding). For 4T1-mScarlet and 67NR-GFP cells, representative cell trajectories in the edge and core compartments, color-coded according to time, are shown (right panels). Scale bars: 100 μm. **i** Δ Distance Index (ΔDI) for 4T1-mScarlet (magenta) and 67NR-GFP (green) cells from spheroids in (**h**). Also see Movies S3–6. A positive ΔDI indicates cell motility towards the spheroid edge ("Out"), and a negative ΔDI indicates movement towards the spheroid center ("In"). Spheroids were treated from day 0 with DMSO control (left) or GM6001 (right). $P = 0.0204$, by the Wilcoxon rank sum test. **j** Mean square displacements (MSDs) for 4T1-mScarlet (magenta) and 67NR-GFP (green) cells from spheroids in (**h**). MSDs were calculated in the radial (left) and angular (right) directions of the polar coordinate system for edge (solid lines) and core (dashed lines) cells in spheroids treated with DMSO (top) and GM6001 (bottom), respectively. The plots are shown in log-log scale, highlighting the super-diffusive ($\alpha > 1$), diffusive ($\alpha = 1$), and sub-diffusive ($\alpha < 1$) nature of motility. Solid lines serve as guides to the eye, indicating the average slopes ($\alpha$ values) corresponding to these different motility modalities. **k** Power law exponent $\alpha_r$ is shown (left) for 4T1-mScarlet (magenta) and 67NR-GFP (green) cells from *edge* and *core* compartments in a spheroid. Effective diffusion coefficient in radial direction is shown for DMSO control (top right) or GM6001 (bottom right) from edge and core compartments in a spheroid.

---

Spheroids of 4T1 cells have been shown to contain laminin, collagen I and fibronectin in the extracellular space[34]. The presence of ECM within a spheroid suggests that 3D cell motility, and consequently cell sorting, may require MMPs. To test the link between motility, MMPs and cell sorting, we treated spheroids with an inhibitor of cell contractility (ROCK inhibitor, Y-27632) or a pan-MMP inhibitor, GM6001. We found that both treatments blocked cell sorting (Fig. 2e, f). We then performed time-lapse imaging of mixed spheroids. We tracked individual cells within the spheroid (Fig. 2h and Movies S3, 4) and for each cell, calculated the difference between the initial and final distance indices (ΔDI = DI for the last position of a given cell – DI for the initial position of a given cell), such that a positive ΔDI indicates cell motility towards the spheroid edge ("out" in Fig. 2g, i), and a negative ΔDI indicates movement towards the spheroid center ("in" in Fig. 2g, i). A null ΔDI indicates no net movement. We once again classified each cell based on their initial position as *edge* or *core*, and defined the edge compartment as a two-cell layer closest to the spheroid-collagen interface (30 μm-wide elliptical ring; Fig. 2g). We found that the percentage of tracked cells that switched between compartments during the duration of the time-lapse imaging was negligible for all cells, except for the 4T1 cells initially located in the core (Fig. S5a). For 4T1 cells, we found that the ΔDI was significantly higher for cells whose initial position was in the spheroid core compared to cells whose initial position was at the spheroid edge (Fig. 2h, i). This indicates that 4T1 cells moved from the spheroid core towards the spheroid edge. In contrast, 4T1 cells located at the spheroid edge had a ΔDI close to zero, suggesting that 4T1 cells that initially found at the spheroid edge moved within that compartment only (Fig. 2i). The ΔDIs were minimal for all 67NR cells, regardless of their initial position, indicating that these cells moved only within the compartments in which they were initially located (Fig. 2i). Since the ΔDI compares the initial and the final positions of cells, this metric does not provide information on the persistence of cells. Indeed, for a given ΔDI, the cell trajectory may be more or less tortuous. By computing the mean square displacement (MSD) of cells in the polar coordinate system, which captures spheroid symmetry, we determined that 4T1 cells are more persistent in the radial direction (super-diffusive, $\alpha > 1$) than 67NR cells (diffusive, $\alpha = 1$) (Figs. 2j, k and S5b–d). Both 4T1 and 67NR cells had similar motility behavior in the angular direction (Figs. 2j and S5b, c). Treatment of spheroids with GM6001 impaired the movement of 4T1 cells from the core to the edge (Fig. 2i–k, Movies S5, 6), and

resulted in a reduced effective diffusion coefficient of all cells in the radial direction (Figs. 2j, k and S5b–d). We confirmed that the spheroid growth was similar in the GM6001 and DMSO conditions, suggesting that the loss of cell sorting was independent from cell proliferation (Fig. S5e). Taken together, these results indicate that cell sorting within mixed spheroids is driven primarily by the directed motility of 4T1 cells from the core to the edge compartment and their ability to remain at the edge, while 67NR cells exhibit random (diffusive) motility, remaining in the same compartment over time. In summary, we showed that differences in persistence drive cell sorting between 4T1 and 67NR cells.

**Cell sorting is E-cadherin independent**. While self-organization and patterning of cells is heavily studied in embryonic tissues and during development, so far, only a few studies tested mixtures of two or more cell types in the cancer spheroid model[19,20]. Most studies reported that cells sort based on the *differential adhesion hypothesis*, which predicts sorting based on differences in the intercellular adhesiveness, with cells that exhibit the strongest cell-cell adhesions positioned in the center[35]. Based on the expression of cadherins for the 4T1/67NR pair, the differential adhesion hypothesis predicts that in a 3D spheroid, E-cadherin-expressing 4T1 will sort from N-cadherin-expressing 67NR cells, with 4T1 cells located in the center and 67NR cells surrounding them. In contrast, our data demonstrates the opposite pattern (Fig. 2a, b), suggesting that the differential adhesion hypothesis may not apply in our model. To confirm this, we inhibited cell-cell junctions in 4T1 cells via E-cadherin blocking antibody (Fig. 3a). Interestingly, blocking E-cadherin did not eliminate cell sorting between 4T1 and 67NR cells, but delayed it by 1 day (Fig. 3b). To confirm this result, we generated stable E-cadherin knock down cell lines (Ecad-KD1 and Ecad-KD2; Fig. 3c–e). According to Western blot analysis, decrease in E-cadherin expression was 36.3% for Ecad-KD1 and 96.8% for Ecad-KD2. Both Ecad-KD cell lines sorted out from the 67NR-GFP cells, and accumulated to the spheroid edge by day 3 post-embedding (Fig. 3g, h). Overall, this suggests that, in our model, cell sorting is independent from E-cadherin.

**An adhesive ECM interface is required for cell sorting**. Since cell sorting occurred by day 3 post-embedding (Fig. 2a, d), and the accumulation of 4T1 cells at the spheroid edge was maintained

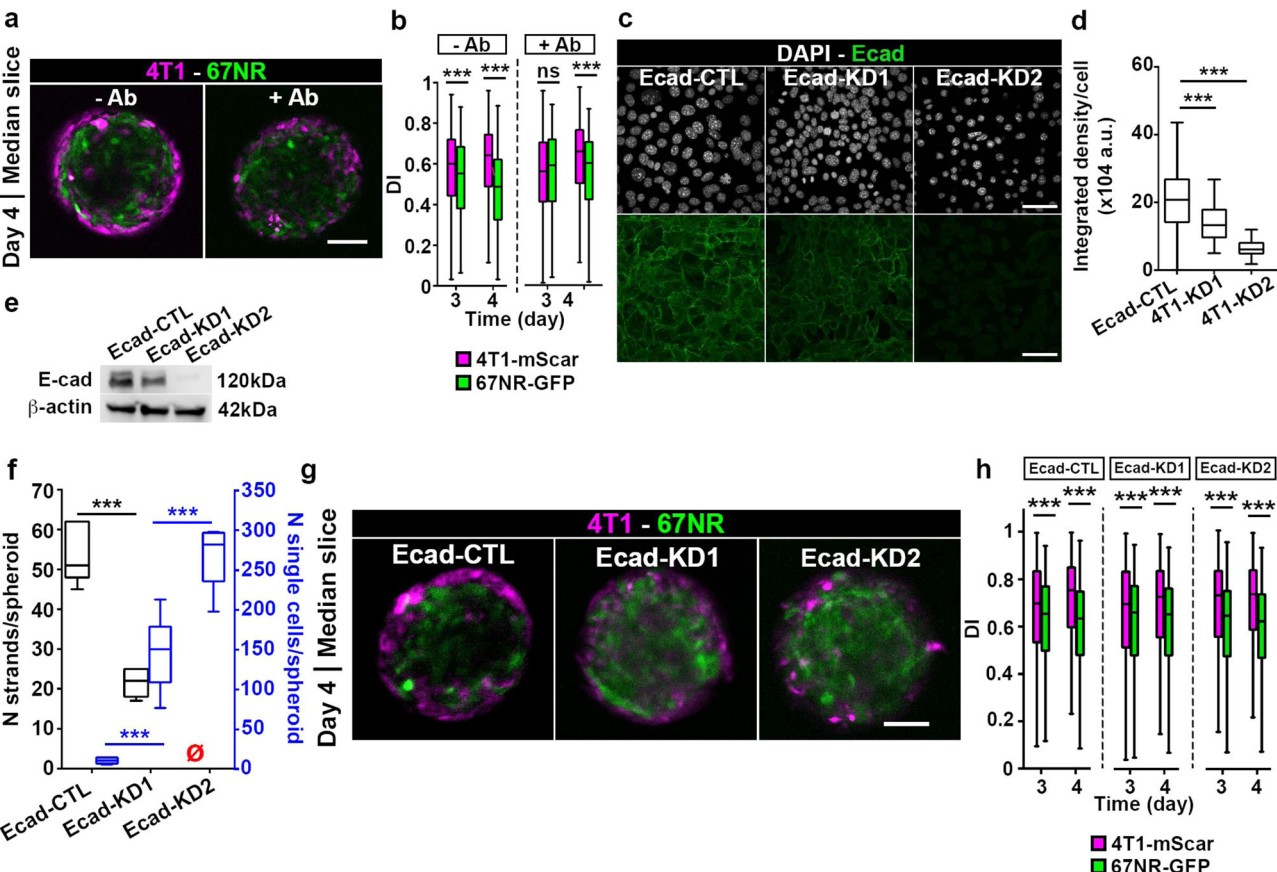

**Fig. 3 Disruption of cell-cell adhesions does not affect cell sorting. a** Mixed spheroids, at a 1:50 ratio, imaged on day 4 post-embedding. Spheroids were treated with (+Ab, right panel) or without (−Ab, left panel) an E-cadherin blocking antibody. Scale bar: 100 μm. **b** DI for 4T1-mScar (magenta boxes) and 67NR-GFP (green boxes) cells from spheroids in (**a**). $P = 2.78 \times 10^{-5}$, $<2.2 \times 10^{-16}$ and $1.12 \times 10^{-12}$ respectively, by the Wilcoxon rank sum test. **c** Ecad-CTL, Ecad-KD1 and -KD2 cells in 2D. E-cadherin (green, bottom panels) and nuclei (gray, top panels) are shown. **d** Integrated density of the E-cadherin signal from cells in (**c**). $P = 4.85 \times 10^{-12}$ and $<2.2 \times 10^{-16}$, by the Wilcoxon rank sum test. **e** Western-blot of E-cadherin expression in Ecad-CTL, -KD1 and -KD2 cells. β-actin is used as a loading control. **f** Number of strands per spheroid (black) and number of single cells per spheroid (blue) for Ecad-CTL, -KD1 or -KD2 spheroids. The red empty symbols indicate zero values. $P = 9.01 \times 10^{-10}$, $1.06 \times 10^{-3}$ and $1.74 \times 10^{-5}$, by the t-test. **g** Mixed spheroids, at a 1:50 ratio, imaged on day 4 post-embedding. Ecad-CTL, -KD1 or -KD2 cells were used. Scale bar: 100 μm. **h** DI for Ecad-CTL, -KD1 and -KD2 (magenta boxes) and 67NR-GFP (green boxes) cells from spheroids in (**g**). $P = 4.72 \times 10^{-5}$, $<2.20 \times 10^{-16}$, $2.21 \times 10^{-5}$, $2.12 \times 10^{-15}$, $2.05 \times 10^{-13}$ and $<2.20 \times 10^{-16}$, by the Wilcoxon rank sum test. Uncropped western blots are available in Supplementary Figs. S14–S15.

throughout 3D invasion (Fig. 2i), we reasoned that the interaction of 4T1 cells with the ECM may be critical for maintaining cell sorting. To test this, we placed mixed spheroids in a non-adherent matrix composed of agarose. At day 3 and 4, the DI was similar for both 4T1 cells and 67NR cells, demonstrating that cells embedded in agarose did not sort (Fig. 4a, b). The area of spheroid core was similar in both collagen I and agarose matrices, indicating that the loss of cell sorting was independent from cell proliferation (Fig. S6a). This absence of cell sorting could be due to the failure of 4T1 cells to remain at the spheroid edge without the adhesive ECM. To test this in real-time, we performed time-lapse imaging of mixed spheroids embedded in agarose and tracked individual 4T1 cells (Fig. 4c; Movie S7). We found that the average ΔDI for cells initially located at the edge compartment was negative, and close to zero for cells initially located in the core compartment (Fig. 4d). This indicates that 4T1 cells move from the spheroid edge compartment into the core compartment (Movie S8), and within the core (Movie S7). Those 4T1 cells, that enter the edge compartment, subsequently leave it, which was not observed in spheroids embedded in collagen I matrix (Movie S9). Interestingly, by computing the MSD of cells in the radial direction, we found that the difference in persistence between 4T1 (super-diffusive, α > 1) and 67NR (diffusive, α = 1) cells was maintained (Figs. 4e

and S6b–d), suggesting that an adhesive ECM interface is required for cell sorting.

This prompted us to hypothesize that, compared to 67NR cells, 4T1 cells preferentially adhere to the ECM. To compare the adhesive properties of 4T1 and 67NR cells to ECM, we developed a 2D cell-ECM adhesive competition assay, in which both cell types were plated on top of circular gelatin islands (5.5 mm in diameter), with poly-L-lysine coating present between the islands (Fig. 4f). After 24 h, we scored the number of 4T1 and 67NR cells that migrated from the gelatin islands onto the poly-L-lysine coated region. We found that 85% of the cells present on the poly-L-lysine region at 24 h timepoint were 67NR cells (Fig. 4g). To explain this, we analyzed the time-lapse movies, and saw that in the rare event of 4T1 cells crossing from the gelatin onto the poly-L-lysine area, cell ended up migrating back to the gelatin (Movie S10). We wondered whether this could be explained by 4T1 cells having a higher adhesion strength to gelatin than 67NR cells. To test this, we measured the contact angle of cells plated on gelatin or poly-L-lysine (Fig. 4h). The cell-matrix contact angle was previously shown to increase with the adhesion strength[36]. We found that both cell types had similar contact angle when plated on gelatin (Fig. 4i). However, 4T1 cells plated on poly-L-lysine had a significantly lower contact angle, while 67NR cells

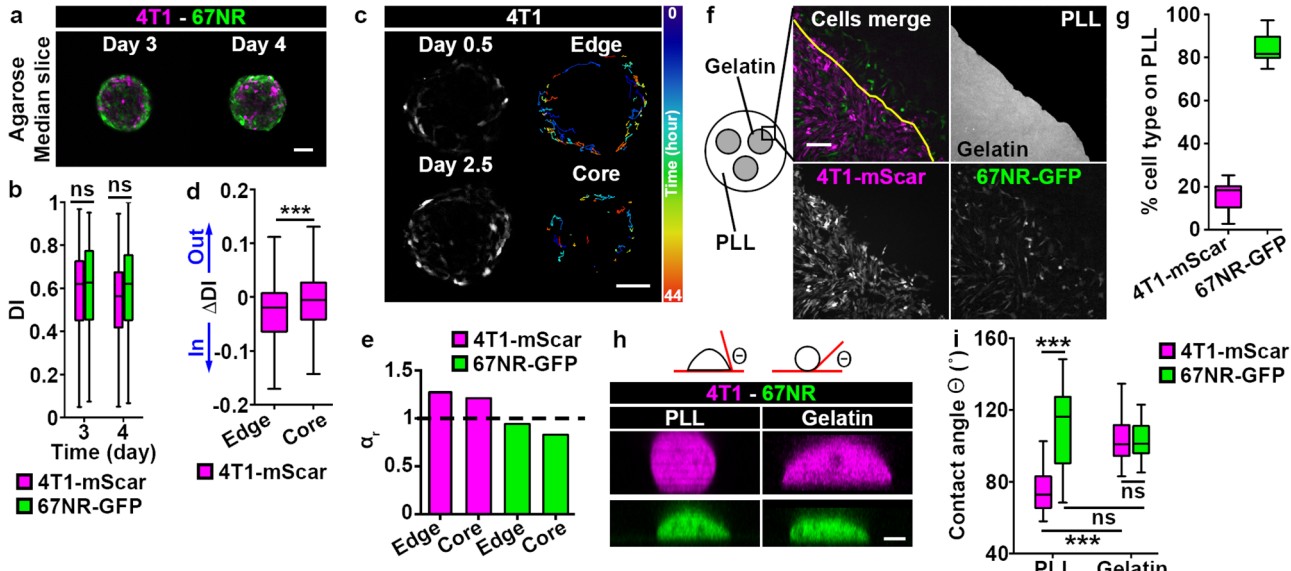

**Fig. 4 4T1, but not 67NR, cells are sensitive to the presence of adhesive ECM. a** Mixed spheroids in agarose matrix imaged on day 3 or 4 post-embedding. Scale bar: 100 μm. **b** DI for 4T1-mScarlet (magenta boxes) and 67NR-GFP (green boxes) cells from spheroids in (**a**). **c** Snapshots of 44 h-long time-lapse recording of a mixed spheroid cultured in a 3D collagen I matrix. Images were taken at 10 h (day 0.5) and 54 h (day 2.5) post-embedding. For clarity, only 4T1-mScarlet cells are shown. Representative cell trajectories, color-coded according to time, are shown (right panels). Also see Movies S7–9. Scale bar: 100 μm. **d** ΔDI for 4T1-mScarlet cells from spheroids in (**c**). $P = 7.50 \times 10^{-4}$, by the Wilcoxon rank sum test. **e** Power law exponent $\alpha_r$ is shown for 4T1-mScarlet (magenta bars) and 67NR-GFP (green bars) cells from *edge* and *core* compartments. **f** Schematic (left) of the 2D cell-ECM competition assay and zoom-in to cells present at the gelatin/poly-L-lysine (PLL) interface 24 h post-plating (top, right panel). 4T1-mScarlet (bottom, left panel) and 67NR-GFP (bottom, right panel) cells were plated on the gelatin islands only. Also see Movie S10. Scale bar: 200 μm. **g** Percentage of 4T1-mScarlet (magenta boxes) and 67NR-GFP (green boxes) cells on poly-L-lysine (PLL) from (**f**). **h** Schematic of the contact angle (θ) (top) and orthogonal, xz views of 4T1-mScarlet and 67NR-GFP cells on poly-L-lysine (PLL) or gelatin, 5 h post-plating. Scale bar: 5 μm. **i** Contact angle θ for cells from (**h**). PLL, 4T1 vs. 67NR: $P = 1.25 \times 10^{-9}$; 4T1, PLL vs. gelatin: $P = 4.13 \times 10^{-10}$, by the Wilcoxon rank sum test.

exhibited similar contact angle values on both gelatin and poly-L-lysine (Fig. 4i, indicating that, compared to 67NR cells, 4T1 cells are more sensitive to the presence of ECM. This affinity to ECM is in line with the higher levels of both FAK and p-FAK observed in 4T1 cells than in 67NR cells (Fig. S6e). By analyzing cell-cell contacts between 4T1 and 67NR cells plated on gelatin, we found that the percentage of homotypic contacts was similar at the time of plating (0 h) and at 24 h post plating (Fig. S6f, g). This confirmed that 4T1 and 67NR cells did not sort in 2D, confirming once more that differential adhesions between cells are not the main driver for cell sorting in our system, and emphasizing the requirement for a cell-ECM interface to initiate cell sorting[35].

**4T1 cells lead 67NR cells in an MMP- and E-cadherin-dependent cooperative invasion**. Following sorting of 4T1 and 67NR cells in spheroids at day 3 and 4, invasion occurred at day 5 and 6 (Figs. 2a, 5a and S4c). One mechanism by which cancer cells can invade collectively is cooperative invasion: invasive leader cells create microtracks inside the ECM, through which non-invasive cells can follow[18–21]. Since 4T1 and 67NR cells display differential invasive skills, we reasoned that 4T1 cells could enable the cooperative invasion of the non-invasive 67NR cells. To test this hypothesis, we analyzed the mixed spheroids at day 6 post embedding. We observed that the non-invasive 67NR cells present in the mixed spheroids entered the collagen I matrix (Fig. 5a). Accordingly, we found that, compared to 67NR-only spheroids, mixed spheroids had a higher number of strands per spheroid (Fig. 5b), and approximately one third of the strands contained 67NR cells (Fig. 5c). Importantly, in mixed spheroids, the majority of strands was led by 4T1 cells (Fig. 5d), which assemble invadopodia (Fig. 1h). In line with the MMP-dependency of 4T1 cells for invasion (Fig. 1a), GM6001 blocked the mixed spheroid invasion

(Fig. 4a–c). During cooperative invasion, cell sorting between 4T1 and 67NR cells was maintained in the invasive strands as well as within the spheroid (Fig. 5e–g), with the 4T1 cells lining the spheroid-matrix interface. Overall, these findings demonstrate that cooperative invasion of mixed spheroids into collagen I is MMP-dependent, with 4T1 cells assuming the leader position.

Our results so far suggest that the 4T1-mediated ECM degradation is required for 67NR cells to enter the collagen I matrix. We hypothesized that 4T1 cells presence in the collagen I matrix is necessary and sufficient to degrade the collagen and create microtracks. To confirm this, instead of pre-mixing 4T1 cells with 67NR cells in a spheroid, we embedded 67NR spheroids in collagen I matrix populated by 4T1 cells. Similar to our observations in mixed spheroids, we found that in the presence of 4T1 cells in the collagen I matrix, 67NR cells invaded into the matrix, and that this invasion was MMP-dependent (Fig. S7a, b). To exclude the possibility that soluble MMPs released by 4T1 cells facilitated the invasion of 67NR cells into the collagen I matrix, we cultured spheroids of 67NR cells with conditioned medium from 4T1 cells. We found that providing conditioned medium from 4T1 cells to 67NR cells was not sufficient for 67NR cells to invade into the collagen I matrix (Fig. S7c, d). In mixed spheroids, we identified microtracks inside the collagen I matrix, filled with 4T1 cells leading and 67NR cells following (Fig. S7e). Previous report of a heterotypic cooperative invasion between cancer-associated fibroblasts and epithelial cancer cells has demonstrated involvement of heterotypic N-cadherin/E-cadherin adhesions[37]. We did not detect heterotypic E-cadherin/N-cadherin junctions between 4T1 and 67NR cells (Fig. S8a–c), suggesting that the cooperative invasion between these cells was not dependent on E- /N-cadherin interactions. Overall, our data suggest that 4T1 cells degrade the collagen I

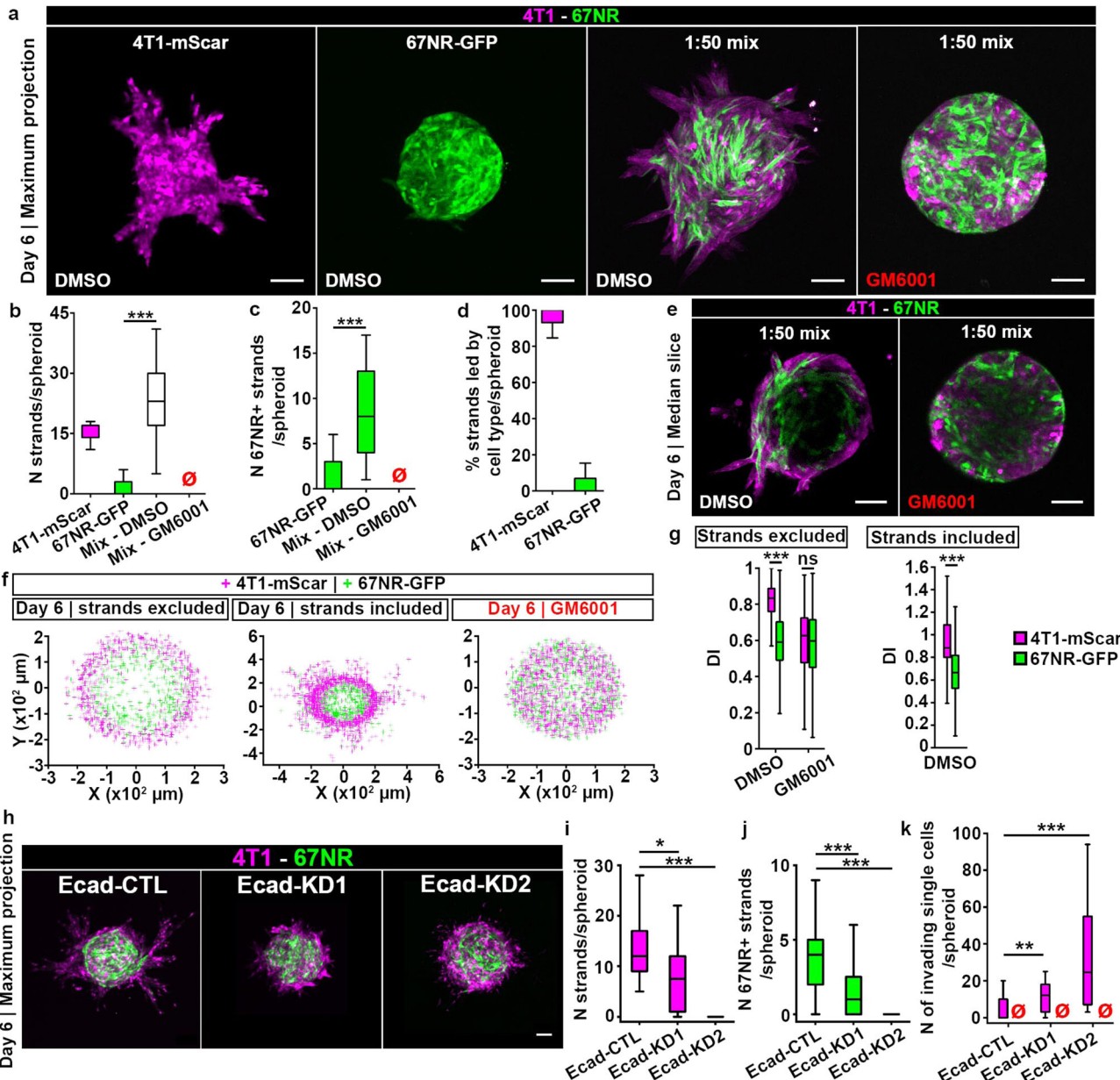

**Fig. 5 4T1 cells lead 67NR cells in an MMP-dependent cooperative invasion. a** Spheroids made of a single or mixed (1:50) 4T1-mScarlet and 67NR-GFP cells, imaged at day 6 post-embedding. Spheroids were treated with DMSO control (left panels) or GM6001 (right panel). Scale bars: 100 μm. Number of strands per spheroid (**b**), the number of strands containing 67NR cells (67NR + ) (**c**), and the percent of strands led by 4T1-mScarlet (magenta boxes) or 67NR-GFP (green boxes) cells (**d**) for single or mixed (white box) spheroids from (**a**). The red empty symbols indicate zero values. $P = 1.90 \times 10^{-7}$ (**b**) and $9.86 \times 10^{-6}$ (**c**), by the Wilcoxon rank sum test. **e** Median slice of mixed spheroids from (**a**). Scale bars: 100 μm. **f** Coordinates of 4T1-mScarlet (magenta) and 67NR-GFP (green crosses) cells from mixed spheroids in (**e**). Cells present in strands were excluded (left panel) or included (middle panel) in the analysis. **g** DIs for 4T1-mScarlet (magenta) and 67NR-GFP (green) cells from spheroids in (**e**, **f**). $P < 2.20 \times 10^{-16}$ and $<2.20 \times 10^{-16}$, by the Wilcoxon rank sum test. **h** Spheroids of Ecad-CTL, -KD1 or -KD2 cells mixed with 67NR at a 1:50 ratio, imaged at day 6 post-embedding. Scale bar: 100 μm. Number of strands (**i**) and number of strands containing 67NR-GFP cells (67NR + ) (**j**) for spheroids from (**h**). $P = 0.02$ and $6.1 \times 10^{-5}$ in (**i**); 0.00073 and $8.70 \times 10^{-8}$ in (**j**), by the Wilcoxon rank sum test. **k** Number of single cells found outside of spheroid core, for spheroids from (**h**). $P = 0.02$ and $6.1 \times 10^{-5}$.

matrix and 67NR cells move into the microtracks created by 4T1 cells.

We wondered if the inhibition of E-cadherin, which had no effect on cell sorting (Fig. 3f, g), can affect cooperative invasion. To test this, we imaged 4T1, Ecad-CTL, -KD1 and -KD2 spheroids on day 6, when invasion occurs (Fig. 5h). While the Ecad-KD1 cell line showed a partial transition from collective to single-cell invasion, with invasive strands as well as single invasive cells, a complete transition to single cell invasion was observed for Ecad-KD2. Similarly, in the mixed spheroids containing 67NR cells with either

Ecad-CTL or -KDs, while Ecad-CTL and Ecad-KD1 cells formed invasive strands, both Ecad-KD cell lines exhibited single cell invasion (Fig. 5i–k). To quantify occurrences of cooperative invasion in both collective and single cell modes of invasion, we measured the number of 67NR followers present either in strands (Fig. 5j), or as single cells (Fig. 5k). Our results demonstrate that while 67NR can follow Ecad-KD1 cells, which maintain invasive strands (Fig. 5i), in the presence of complete transition to single cell invasion, such as in mixed spheroids of Ecad-KD2 and 67NR, cooperative invasion is lost (Fig. 5k).

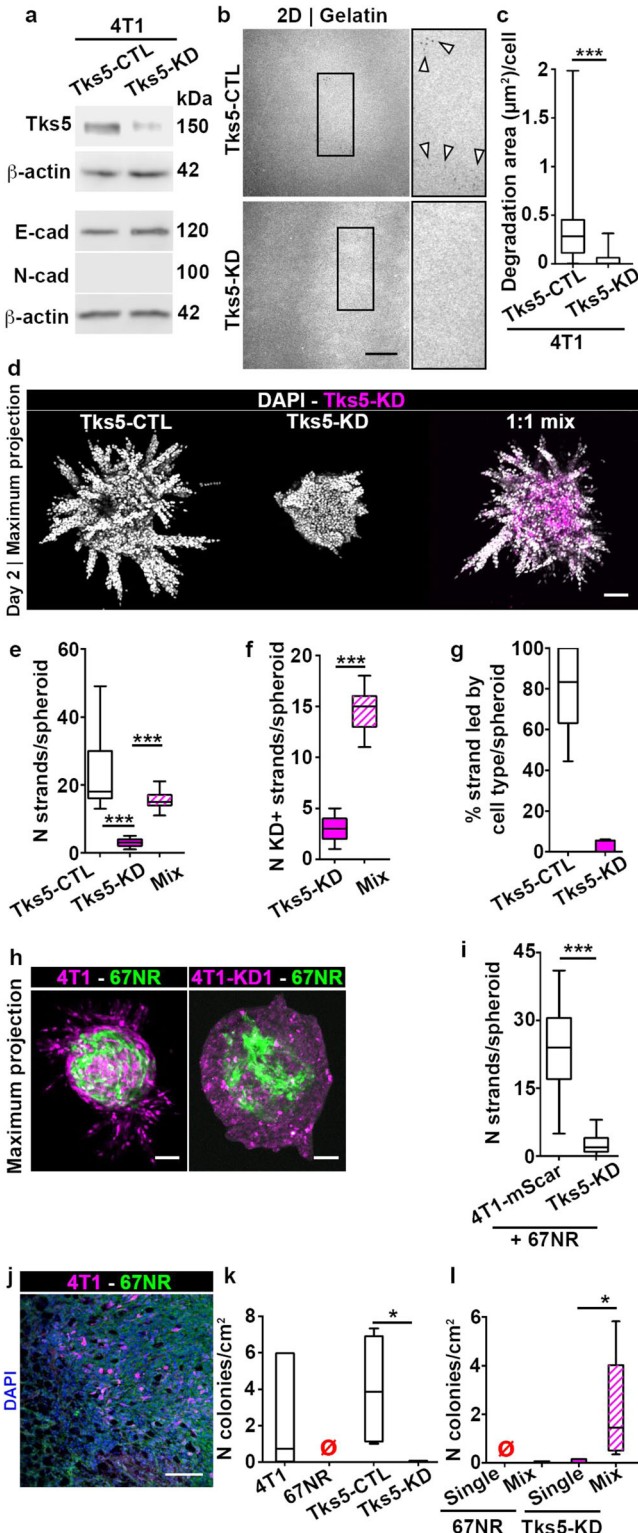

**Fig. 6 Cells without invadopodia can invade and metastasize via cooperation with cells that assemble invadopodia. a** Tks5 (top), and E/N-cadherin (bottom) expression in Tks5-CTL and Tks5-KD cells. **b** Gelatin degradation by Tks5-CTL (top panel) and Tks5-KD (bottom panel), 18 h after plating. The inserts show a 2X zoom-in of the boxed area and arrowheads indicate representative degradation. Scale bar: 20 μm. **c** Degradation area per cell for Tks5-CTL and Tks5-KD cells from (**b**). $P < 2.2 \times 10^{-16}$, by the Wilcoxon rank sum test. **d** Day 2 images of spheroids made of Tks5-CTL, -KD or a mixture of Tks5-CTL and -KD (1:1 ratio) cells. Scale bar: 100 μm. Number of strands per spheroid (**e**), number of strands containing Tks5-KD cells (**f**), and percentage of strands led by Tks5-CTL and -KD cells (**g**) in spheroids from (**d**). $P = 1.54 \times 10^{-7}$ and $2.89 \times 10^{-5}$, by the Wilcoxon rank sum test in (**e**). $P = 6.69 \times 10^{-11}$, by the t-test in (**f**). **h** Day 6 images of mixed spheroids (1:50 ratio) made of 67NR-GFP and 4T1-mScarlet (4T1-mScar) or Tks5-KD cells. Scale bars: 100 μm. **i** Number of strands per spheroid from (**h**). $P = 3.36 \times 10^{-6}$, by the Wilcoxon rank sum test. **j** Mixed 4T1-mScarlet and 67NR-GFP tumor, labeled with DAPI (blue). Scale bar: 50 μm. **k** Number of lung colonies per cm$^2$ for mice inoculated with 4T1, 67NR, Tks5-CTL or Tks5-KD cells. The red empty symbols indicate zero values. $P = 0.032$, by the Wilcoxon rank sum test. **l** Number of lung colonies per cm² for mice inoculated with single 67NR cells, or mixed with 4T1 cells; and single Tks5-KD cells, or mixed with Tks5-CTL cells. The red empty symbols indicate zero values. $P = 0.032$, by the Wilcoxon rank sum test. Uncropped western blots are available in Supplementary Figs. S16–S18.

established the corresponding control (Tks5-CTL) cell line (Fig. 6a, top, 78.1% knockdown efficiency). We also verified that Tks5-CTL and Tks5-KD cell lines still expressed E-cadherin (Fig. 6a, bottom), which was localized at the junctions between Tks5-CTL and Tks5-KD cells in spheroids (Fig. S9a, b). We confirmed by gelatin degradation assay in 2D and spheroid invasion assay in 3D that Tks5-KD cells lost their invasive capacity (Fig. 6b–e). We then performed the spheroid invasion assay of mixed spheroid containing Tks5-CTL and Tks5-KD cells. At day 2 post-embedding, we found that both Tks5-CTL cells and the non-invasive Tks5-KD cells had entered the collagen I matrix (Fig. 6d). Mixed spheroids had a similar number of strands to Tks5-CTL (Fig. 6e), and most strands contained both cell types (Fig. 6f) and were led by Tks5-CTL cells (Fig. 6g). To test if cooperative invasion was cell type specific, we chose the meta-static human cell line MDA-MB-231, which does not express E-cadherin and assembles functional invadopodia[13]. By knocking down human Tks5 in MDA-MB-231 cells (Fig. S10a–c), we observed that cells without invadopodia were able to follow cells with invadopodia, via cooperative invasion (Fig. S10d–g), strengthening our results. To further test the importance of invadopodia in cooperative invasion of cancer cells, we generated mixed Tks5-KD and 67NR-GFP spheroids. Here, we did not observe invasive strands (Fig. 6h, i). Finally, we validated the conclusion with a different Tks5 shRNA sequence, Tks5-KD2, which provided results similar to Tks5-KD (Fig. S11). Overall, we demonstrate that functional invadopodia are required in leader cells during cooperative invasion, while follower cells can lack invadopodia.

We also tested if the removal of invadopodia affected cell sorting. As expected, cell sorting did not occur in mixed spheroids of Tks5-CTL and Tks5-KD cells (Fig. S12a), but it did occur in spheroids containing Tks5-KD and 67NR-GFP cells (Fig. S12b). These findings confirmed that invadopodia are not required for cell sorting. Since Tks5-CTL and Tks5-KD cells display similar motility and cell-ECM adhesion properties (Fig. S12c, d), these observations also strengthen the requirement for differential motility and differential cell-ECM sensitivity for cell sorting to

**Cells with invadopodia lead cells without invadopodia in cooperative invasion.** Given that 4T1-mediated ECM degradation is required for cooperative invasion (Figs. 4 and S6), and that 4T1 leader cell assemble invadopodia (Fig. 1h), we wondered whether specifically invadopodia assembly by 4T1 cells was responsible for ECM degradation. To rigorously confirm that invadopodia function is required for cooperative invasion of cancer cells, we stably eliminated invadopodia in 4T1-mScarlet cells, using a knockdown of mouse Tks5 (Tks5-KD)[13], and

occur. In accordance with this, cell sorting did not occur in spheroids containing 4T1-mScarlet and 4T1 wild type cells (Fig. S12e, f).

**Mixtures of cells with invadopodia and without invadopodia perform cooperative metastasis.** Dissemination and metastasis require transendothelial migration of cancer cells, i.e. intravasation and extravasation, which were previously shown to be invadopodia-dependent[13,38]. While interstitial ECM in the primary tumor and surrounding tissues may be permanently remodeled by microtracks formation[39], transendothelial migration involves brief, transient opening of perivascular ECM[40]. It is not known whether invasive and non-invasive cells may also cooperate during metastasis. To investigate whether cells with invadopodia could enable metastasis of cells without invadopodia, we generated tumors with single or mixed cell lines. After tumors reached 6–9 mm in diameter, we performed the lung clonogenic assay on digested tissues. To determine which cell type was growing metastatic lung colonies, we leveraged the differences in fluorescent protein expression or drug sensitivity. We confirmed that the Tks5 knock down was maintained in the Tks5-KD tumors (Fig. S11i). We found that the Tks5-KD cell lines, in 4T1 or MDA-MB-231 background, and 67NR cells were not capable of lung metastasis, in contrast to control and wild type cells (Figs. 6k and S10i, S11j). When 4T1 and 67NR cells were co-injected, 67NR did not metastasize (Fig. 6l). This was not due to 4T1 cells taking over as 67NR cells were present in the tumor tissue (Fig. 6j). Similarly, the co-injection of MDA-MB-231 control and its corresponding Tks5 knockdown, D2-KD, demonstrated that only control cells were capable of metastasis (Fig. S10h, i). Interestingly, when Tks5-KD were co-injected with Tks5-CTL cells, both cell types were observed in the lungs (Fig. 6l), suggesting that cooperative metastasis has occurred. Collectively, these results imply the dependence of cooperative metastasis on invadopodia and on E-cadherin expression and/or presence of cell-cell junctions.

## Discussion
In this study, we show that invasive cells can sort from, and lead non-invasive cells during cooperative invasion and metastasis. By examining cell movements in spheroids, we demonstrate that differential persistence and ECM sensitivity drive cell sorting between clones. Through invadopodia removal, by silencing Tks5, we demonstrate that cells with invadopodia lead cells without invadopodia in cooperative invasion, and enable their metastasis.

We confirm previously reported observations that the invasive 4T1 cells are in a hybrid epithelial/mesenchymal state, while the non-invasive 67NR cells are mesenchymal[27–29]. It was previously shown that the expression of Twist, which is necessary for invadopodia[8], is present in 4T1 cells but not in 67NR cells[41]. Hence, Twist-mediated EMT could be responsible for the differences in invasion skills between 4T1 and 67NR cells[41]. Since 4T1 cells but not 67NR cells assemble functional invadopodia, it seems that specific invadopodia-granting EMT trajectory, rather than EMT completion is required for invadopodia emergence. Our results are in line with the recent evidence that epithelial cells[3,42] and hybrid epithelial/mesenchymal cells can metastasize, sometimes more efficiently than mesenchymal cells[43,44].

Despite expressing the core invadopodia components cortactin and Tks5, 67NR cells fail to degrade the matrix. However, 67NR cells lack active MMP-2 and MMP-9, likely due to MMP-14 (also known as MT1-MMP), protease commonly in charge of activating MMP-2 and MMP-9, is not being functional, or not being delivered to the plasma membrane[45,46].

Here, we uncover the role of differential persistence and cell-ECM sensitivity in establishment of cell sorting in cancer. Classical studies on cell sorting during development showed cell sorting commonly relies on differential strength of cell-cell adhesions[35], or on differences in contractility[47]. In contradiction to these views, we show that 4T1 and 67NR cells did not sort on a 2D gelatin layer, or when placed in a 3D agarose matrix. The importance of differential motility in sorting was previously suggested during tissue patterning[48], and in engineered breast tubules, where it was shown that more directional epithelial cells accumulate at the tissue edge[49]. During self-organization of mammary ducts, presence of binary cell-ECM interactions (on or off) was reported to regulate cell sorting[36]. Similar to our observations, Pawlizak et al. recently demonstrated that sorting of breast cell lines expressing E-, N- or P-cadherin could not be explained by the differential adhesion hypothesis[50]. Interestingly, the authors proposed that motility could be responsible for cell sorting.

We find that in the spheroids where highly-persistent, invasive cells sensitive to ECM are mixed with non-invasive, randomly moving cells, sorting and accumulation of invasive cells at the spheroid-ECM interface precedes cooperative invasion. This emphasizes the importance of studying the mechanisms regulating the spatial organization of leader and follower cells. The accumulation of leader cells at the spheroid-matrix interface may enhance the speed and efficiency of the cooperative invasion. Supporting these views, a recent study demonstrated that cell sorting precedes the basal extrusion of mammary epithelial cells[51]. Importantly, while cell sorting may be catalytic, it does not seem to be essential for the cooperative metastasis in our system. While mixed 4T1 Tks5-KD and Tks5-CTL cells engage in cooperative metastasis, but do not sort, mixes of 4T1 and 67NR cells sort but do not metastasize cooperatively.

Since the discovery of cooperative invasion, significant work has been carried out to uncover the mechanisms by which heterogeneous breast cancer cell populations interact and mobilize collectively. Our work suggests that invadopodia activity is a determinant of the leader cell phenotype. We previously showed that the G1 phase of the cell cycle is also a determinant of the leader cell identity[22]. Consistent with this present work, we had also proved that invadopodia are enriched in the G1 phase of the cell cycle[22]. A different study revealed that cells leading invasion strands possess higher intracellular energy compared to follower cells[17]. Altogether, it seems that the cell cycle, intracellular energy and invadopodia function are determinants of the leader cell identity. However, the interplay between the cell cycle, intracellular energy and invadopodia function has yet to be investigated in the context of the emergence of leaders and followers within a cell population.

In summary, our study suggests that cooperativity between cancer clones may be an efficient mechanism for collective metastasis. Specifically, we demonstrate that invadopodia enable cooperative metastasis and allow non-invasive cells to metastasize. Our findings on cooperative metastasis of 4T1 Tks5-KD with Tks5-CTL are in alignment with the previous demonstration of dissemination of epithelial cell clusters[3]. In contrast, 67NR cells do not cooperatively metastasize when mixed with 4T1. In addition, MDA-MB-231 Tks5-KD (D2-KD) cells do not cooperatively metastasize with MDA-MB-231 cells. Both of these cell lines lack E-cadherin, suggesting that the cooperative metastasis is dependent on strong E-cadherin-based cell-cell interactions. Interestingly, cooperative invasion in 3D spheroid assay does not pose such a requirement, as both 67NR and D2-KD cells successfully follow their invasive counterparts. The likely reason is that the collagen deformations generated by the invasive leader cells are plastic (permanent), allowing either cell strands or

individual cells to migrate through them[39]. In contrast, deformations that leader cells generate to the blood vessel wall are elastic (transient), and followers can cross blood vessel walls only if they exhibit strong cell-cell adhesions to the leader.

Our findings demonstrate that cancer cells engage in cooperative metastasis, which may be more detrimental than metastasis of single cells. We propose that targeting invadopodia could be a potent strategy to inhibit metastasis of both individually[13,14], as well as collectively invading cells.

## Methods

**Ethics statement**. All experiments on mice (*Mus musculus*) were conducted in accordance with the NIH regulations and approved by the Temple University IACUC protocol number 4766.

**Fabrication of the spheroid imaging devices (SIDs)**. Spheroid imaging devices (SIDs) were fabricated as previously described[33]. Briefly, SIDs were made by binding poly-dimethyl-siloxane (PDMS) disks to the glass-bottom dishes (MatTek Corporation). Each PDMS disk measures 17.5 mm in diameter and contains three wells, 5.5 mm in diameter, suited for individual spheroids.

**Gelatin coating of 6-well plates**. 6-well plates were coated with gelatin as previously described[52]. Briefly, each well was coated with a 2.5% gelatin solution for 10 min followed by treatment with 0.5% glutaraldehyde (Sigma-Aldrich) for 10 min, on ice, and an extra 30 min at room temperature. Plates were sterilized by 70% ethanol and then 50 U/mL penicillin - 50 μg/mL streptomycin treatment.

**Cell culture**. The isogenic murine breast cancer cell lines 4T1 and 67NR were gifts from Dr. Fred R. Miller at the Karmanos Cancer Center and Dr. Jin Yang from UCSD. The human breast cancer cell line MDA-MB-231 (HTB-26) was obtained from the American Type Culture Collection. The MDA-MB-231-Dendra2-hTks5 KD cell line was described previously and was kept under continuous 0.5 μg/ml puromycin and 500 μg/ml geneticin pressure[13]. All cells were cultured in Dulbecco's modified eagle medium [4.5 g/L D-glucose, L-glutamine] (DMEM, Gibco), supplemented with 10% fetal bovine serum (FBS, Atlanta Biologicals) and 50 U/mL penicillin - 50 μg/mL streptomycin (Gibco). Cell cultures were maintained at 37 °C and 5% $CO_2$ for a maximum of 60 days.

**Plasmid transfections**. The 4T1-mScarlet and 67NR-GFP cell lines were generated by transfection using a 1:4 ratio of plasmid DNA:FugeneHD reagent (Promega), according to the manufacturer's instructions, followed by selection with 500 μg/mL geneticin (Fisher BioReagents) and 3 μg/mL puromycin (MP Biomedicals), respectively. pmScarlet-H-C1 was a gift from Dorus Gadella (Addgene plasmid # 85043). pEGFP-puro was a gift from Michael McVoy (Addgene plasmid # 45561). The MDA-MB-231-mScarlet cell line was generated by electroporation (Lonza) of the pmScarlet-C1 plasmid, according to the manufacturer's instructions, and selection with 500 μg/ml geneticin. After two weeks of drug selection, cells were sorted (BD FACSAriaIIμ, BD Biosciences), collecting the subpopulations expressing high levels of mScarlet or GFP.

**Lentivirus transduction and testing KD efficiencies**. The knockdown cell lines Tks5-KD and -KD2, E-cadherin-KD1 and -KD2, and the knockdown control cell lines Tks5-CTL, Ecad-CTL and MDA-MB-231-mScarlet-CTL were generated by transduction of the 4T1-mScarlet, 4T1 and MDA-MB-231-mScarlet cell lines, respectively, with lentiviral particles (3 viral particles/cell) containing shRNA targeting mTks5 (Clone ID: TRCN0000105733, CGTGGTGGTGTCCAACTATAA; Clone ID: TRCN0000105734, CCTCATACATTGACAAGCGCA), hTks5 described previously[13], or E-cadherin (KD) (Clone ID: TRCN0000042581, CCGAGAGAGTTACCCTACATA; Clone ID: TRCN0000042579, CGGGACAATGTGTATTACTAT) or non-targeting shRNA (CTL) in the pLKO.1-puro vector (MISSION library, Sigma-Aldrich), and selection with 2 μg/mL puromycin 3-7 days after infection. Western blots were analyzed to confirm KD efficiencies. In addition, images of Ecad-CTL and Ecad-KDs immunolabeled with E-cadherin and segmented using Cellpose. Masks were overlaid on the image and integral density per cell was quantified.

**2D proliferation assay**. Crystal violet staining was used to assess the effect of mitomycin C on the proliferation of the 4T1 cell line. Briefly, 4T1 cells were seeded in a 6-well plate and the next day, the culture medium was replaced with culture medium containing 0.5 μg/mL Mitomycin C (resuspended in DMSO, Cayman Chemical). After 2 days, cells were washed with cold PBS, fixed with ice-cold 100% methanol for 10 min and stained with 0.5% crystal violet solution in 25% methanol for 10 min at room temperature. Excess dye was removed by several washes with tap water and the plate was air dried overnight at room temperature. The dye was solubilized using 100% methanol for 20 min and the optical density was read on a plate reader at 570 nm. The optical density at 570 nm for mitomycin C-treated cells was reported to the optical density at 570 nm for DMSO-treated cells.

**2D gelatin degradation assay**. Gelatin was fluorescently labeled with Alexa-405-NHS ester and 35 mm glass bottom dishes (MatTek Corporation) were coated with Alexa-405-gelatin as previously described[31]. 400,000 (4T1/67NR) or 300,000 (MDA-MB-231) cells were plated per dish and cells were fixed 18 h later with 4% paraformaldehyde (Alfa Aesar) for 10 min, permeabilized with 0.1% Triton X-100 (Calbiochem) for 5 min, blocked with 1% FBS/1% BSA (Sigma-Aldrich) in PBS (Gibco) for 3 h, incubated with anti Tks5 antibody (Millipore, MABT336) for 2 h, then with secondary antibody and Alexa Fluor 633 Phalloidin (Invitrogen) for 1 h.

Samples were imaged on a laser scanning confocal microscope (FV1200, Olympus) using a 60X objective (UPLSAPO60XS, 1.35 NA, Olympus). Stacks were collected at 1 μm z-step. To quantify matrix degradation, images were processed using a custom macro in Fiji. Briefly, slices from the stack were z-projected using the *Max Intensity* method, followed by thresholding of the signal in the gelatin channel, using the *Automatic Threshold* algorithm, and measuring the area of degradation spots using the *Particle Analysis* tool. To account for the differences in the cell density across fields of view, the total area of degradation in a field of view was divided by the total number of cell present in this field of view. Cells were counted using the F-actin staining.

**Scratch assay**. 6-well plates were coated with 50 μg/ml poly-L-lysine (Sigma-Aldrich) for 20 min and air-dried. Cells were plated and cultured to confluency before a 10 μl pipet tip was used to create a cross-shaped wound across the monolayer. Samples were imaged on a widefield microscope (IX-81, Olympus) equipped with an LED lamp (Excelitas Technologies), an Orca 16-bit charge-coupled device camera (Hamamatsu), an automated z-drift compensation IX3-ZDC (Olympus), an automated stage (Prior Scientific), an environmental chamber (Live Cell Instrument) and using a 10X objective (MPlanFL N 10X, 0.3 NA, Olympus). Cell motility was recorded at 10 min intervals over 48 h. Manual cell tracking was performed using the TrackMate plugin through Fiji[53]. The track number, the spot coordinates and the frame number were exported. Computation of the velocity and persistence were done using a custom made Matlab code.

**Gelatin islands and 2D cell-ECM adhesive competition assay**. To generate gelatin islands, we utilized the PDMS inserts (see *Fabrication of the spheroid imaging devices (SIDs)*). Briefly, 35 mm glass bottom dishes (MatTek Corporation) were coated with 50 μg/ml poly-L-lysine (Sigma-Aldrich) for 20 min and air-dried. Then, PDMS rings were gently placed on top of the glass and sealed by gently pressing down. Next, each 5.5 mm-diameter hole was coated with fluorescently labeled gelatin as previously described[31] (see *2D gelatin degradation assay*). 4T1-mScarlet and 67NR-GFP cells were plated in each hole at a 1 to 1 ratio. Cells were allowed to adhere for 1 h, after which the PDMS inserts were gently peeled off the glass and medium was added to the dishes. Finally, cells were fixed 24 h later with 4% paraformaldehyde (Alfa Aesar) for 10 min.

Samples were imaged on a widefield microscope (Eclipse Ti2-E, Nikon) equipped with a pco.panda sCMOS camera (PCO) and using a 10X objective (CFI Plan Fluor 10X, 0.3 NA, Nikon). Tiles (6 × 6) were acquired to visualize the entire gelatin island surface. Live cells were imaged every 10 min and using an environmental chamber (Tokai Hit). The movie was annotated using the DrawArrowInMovie Fiji plugin[54]. To quantify the 2D cell-ECM adhesive competition assay, we counted the number of 4T1 and 67NR cells that migrated off the gelatin islands. To quantify cell sorting in 2D, we only utilized regions coated with gelatin and we counted the number of homotypic neighbors.

**Measurements of the cell-ECM contact angle**. Measurement of the contact angle was performed as previously described[36]. Briefly, 35 mm glass bottom dishes (MatTek Corporation) were coated with fluorescently labeled gelatin as previously described[31] or with 50 μg/ml poly-L-lysine (Sigma-Aldrich) for 20 min and let to air dry. Cells were left to adhere for 5 h before imaging. Solitary cells, which had no physical interaction with nearby cells, were imaged on a laser scanning confocal microscope (FV1200, Olympus) using a 60X objective (UPLSAPO60XS, 1.35 NA, Olympus) equipped with an environmental chamber (In Vivo Scientific), with a 1 μm z-step. *Orthogonal Views* was used to measure the angle between the ECM and the main body of the cell: the contact angle.

**3D spheroid invasion assays**. For 4T1 and 67NR cell lines, 3D spheroids were generated by the hanging drop method. 3000 cells per 40 μl drop containing 4.8 mg/mL methylcellulose (Sigma-Aldrich), 20 μg/mL Nutragen (Advanced Biomatrix), were placed on the lid of tissue culture dishes. The lids were carefully turned and placed on the bottom reservoir of the dishes filled with PBS to prevent evaporation. Alternatively, for MDA-MB-231 cell lines, 3D spheroids were generated in a 96-well V-bottom dish coated with 0.5% poly(2-hydroxyethyl methacrylate) (Sigma-Aldrich) in ethanol. Then, 5000 cells in 50 μl of medium were distributed to each well and the plate was centrifuged for 20 min at 1000 × *g* and at 4 °C. Finally, 50 μl of Matrigel (Corning) was added to each well at a final concentration of 2.5%. The spheroids were formed over 3 days at 37 °C and 5% $CO_2$, embedded in 30 μl of 5 mg/mL rat tail Collagen I (Corning, alternate gelation

protocol) and placed into the SIDs. Collagen I was polymerized at 37 °C for 30 min and then culture medium was added to the dishes. For drug treatments, cell culture medium containing 25 µM GM6001 (resuspended in DMSO, Cayman Chemical), 10 µM Y-27632 (resuspended in DMSO, Cayman Chemical) or 0.1% DMSO control was used.

For the experiments where 4T1-mScarlet cells surround the 67NR-GFP spheroids, 4T1-mScarlet cells were added to the collagen mix containing the 67NR-GFP spheroids at $10^6$ cells/mL before polymerization of the collagen.

For the experiments where conditioned medium was used, 4T1-mScarlet cells were seeded onto a gelatin-coated 6-well plate, at $2 × 10^6$ cells/well. 2 mL of complete DMEM were used per well and the embedded 67NR-GFP spheroids were cultured, from day 0, using the conditioned medium from the 4T1-mScarlet cells plated onto gelatin. Every two days, the conditioned medium was replaced.

Collagen was labeled as previously described[55], using 2 µg/ml of 405-Alexa-NHS ester (Biotium).

To block E-cadherin, 5 µg/ml of the blocking antibody (MABT26, Millipore) was used to disrupt cell-cell adhesions. Ecad-KD spheroids were generated as previously described[19]. Briefly, cells were trypsinized, and resuspended at $1.5 × 10^4$ cells/ml in complete DMEM F12 medium (5% horse serum, 0.5 µg/ml hydrocortisone, 20 ng/ml hEGF, 10 µg/ml insulin, 100 ng/ml cholera toxin, 1% penicillin/streptomycin) and 0.25% methylcellulose (Sigma-Aldrich). Cell suspension was seeded into non-adhesive, round-bottom 96-well plates (Corning), 200 µl/well. The plate was centrifuged at 1000 rpm for 5 min at room temperature and placed on the orbital shaker at 37 °C, 5% CO2 for 2 h. The medium was replaced with a complete DMEM F12 containing 0.25% methylcellulose (Sigma-Aldrich) and 1% Matrigel (Corning). Spheroids were formed in the incubator for 48 h.

**Immunolabeling, imaging and analysis of fixed spheroids**. Immunofluorescence labeling was performed as previously described[33]. Briefly, the embedded spheroids were simultaneously fixed and permeabilized in 4% paraformaldehyde and 0.5% Triton X-100 in PBS for 5 min, further fixed in 4% paraformaldehyde in PBS for 20 min and blocked in 1% FBS/1% BSA in PBS at 4 °C for 24 h on a shaker. The embedded spheroids were then incubated with the anti-collagen I ¾ (immunoGlobe, 0207-050; 1 to 100), anti E-cadherin (Invitrogen, 13-1900; 1 to 100), anti N-cadherin (BD Transduction Laboratories, 610920; 1 to 100) and anti Tks5 (Millipore, MABT336; 1 to 100) overnight at 4 °C and with secondary antibodies and Alexa Fluor 633 Phalloidin (Invitrogen; 1 to 250) for 1 h at room temperature on a shaker.

Spheroids were imaged using a laser scanning confocal microscope (FV1200, Olympus) with a 10X (UPLXAPO10X, 0.4 NA, Olympus) or a 30X objective (UPLSAPO30XSIR, 1.05 NA, Olympus), using a 3–5 µm z-step. To quantify the total spheroid area, spheroids were labeled with DAPI. The images were processed using a custom macro in Fiji. Briefly, slices were z-projected using the *Max Intensity* method and the nuclei were selected using the *Automatic Threshold* algorithm from Fiji. Then, the *Particle Analysis* tool was used to measure the total area of nuclei.

To quantify the E-cadherin signal, the slices of interest were z-projected using the *Max Intensity* method and strands or single cells were identified. For the relative junction/cytosol ratio: a 10 µm-long line was drawn across and centered at the junction between two cells within a strand. The gray value along the line was measured for the E-cadherin and F-actin channel using the *Plot Profile* tool. The gray value at 0 and 5 µm were defined as the "cytosol" and "junction" signal, respectively.

To quantify cell sorting, images were processed using a custom macro in Fiji. Since maximum projection of the z-slices introduces artefacts in the positions of cells, regarding the spheroid edge vs. core compartments, we utilized the median slice of the z-stack only. Briefly, the median slice was extracted from the z-stack and the spheroid core was selected in the brightfield channel using the *Automatic Threshold* algorithm from Fiji. Then, using the *Fit Ellipse* and *Centroid* options in *Measurements*, the coordinates of the spheroid core center and the major axis of the spheroid core were extracted. Finally, the *Multi-point* tool was used to record the coordinates of GFP + and mScarlet+ cells. Cell sorting was quantified as the distance from the spheroid center to the cell (d in Fig. 2c) over the semi major axis of the spheroid (a in Fig. 2c). We defined this ratio as the "Distance Index" (DI). Alternatively, for mixed spheroids that contained non-labeled cells, like Tks5-CTL (Fig. 6) or 4T1 wild type cells (Fig. S12), DAPI staining was used to measure a cell's coordinate. For a given tracked cell, we defined the ΔDI as the DI for its final position minus the DI for its initial position.

**Live imaging of spheroids and image processing**. Live imaging of spheroids was performed either longitudinally (daily), to analyze cell sorting, or via time-lapse, to analyze cell motility. Confocal microscope (FV1200, Olympus) using a 10X objective (UPLXAPO10X, 0.4 NA, Olympus) equipped with an environmental chamber (Live Cell instrument) was used. Cell motility was recorded at 10- or 20-min intervals over 44 h, with a 15 µm z-step. Only slices where both the cells from the core and the edge compartments were visible, were used for tracking. Cell tracking was performed using the TrackMate plugin through Fiji[53]. Spot detection was done using the LoG detector with median filtering and subpixel localization. Then, the linear motion LAP tracker was used to link spots. Tracks were filtered

based on the number of spots in the track with a ≥ 5 spots/track cutoff. Tracks were visually validated and corrected if needed, using the TrackScheme tool. Finally, gaps in tracks were closed by introducing new spots. The new spots position was calculated using linear interpolation. For each track, the distance from the original spot to the spheroid-collagen I interface was measured and if this distance was ≤30 µm, the track was classified as *edge*, otherwise the track was classified as *core*. The track number, the spot coordinates and the frame number were exported. Computation of the distance index was done using a custom made Matlab code (available upon request).

**Mean-squared displacement analysis of cells**. Coordinates of individual cell trajectories were obtained by particle tracking with the origin of each spheroid at (0,0), and were transformed from the Cartesian {x,y} to {r,ϕ} polar coordinate system. Mean squared displacements (MSDs) were then calculated in radial and angular directions, averaged over different spheroids and experimental repeats for different conditions. The MSD data were then fit to a power law, in both directions, namely

$$\left\langle (r - r_0)^2 \right\rangle = \Gamma_r t^{\alpha_r} \tag{1}$$

and

$$\left\langle (\phi - \phi_0)^2 \right\rangle = \Gamma_\phi t^{\alpha_\phi} \tag{2}$$

where $(r_0, \phi_0)$ correspond to the origin of each trajectory in the polar coordinate system, $\Gamma_{r,\phi}$ correspond to the amplitudes of the power laws, and $\alpha_{r,\phi}$ are the exponents, in radial and angular directions, respectively. MSD data were fit to these power laws using Levenberg-Marquart algorithm, with the range of fits in the interval [0, 3 h]. Note that the motility of cells is sub-diffusive for α < 1, diffusive for α = 1, and super-diffusive for α > 1.

In the radial direction, since the motion was found to be super-diffusive, we also calculated the time-dependent effective diffusion coefficient given by[56]

$$D_{eff}(t) = (\Gamma/2)t^{\alpha-1} \tag{3}$$

resulting in

$$\left\langle (r - r_0)^2 \right\rangle = 2\,D_{eff}(t)t \tag{4}$$

It is important to note that, for motion that is not diffusive, time-dependent $D_{eff}(t)$ is the only way to estimate a diffusion coefficient with the correct units, and compare different data sets in a consistent way, as long as the same time points are chosen for comparison.

**Western blot assay**. Cells were plated onto poly-L-lysine coated dishes and cultured to 80% confluency. Cells were harvested in ice-cold RIPA lysis buffer (Teknova), supplemented with protease inhibitors (complete cocktail, Roche) and phosphatase inhibitors (Halt cocktail, Sigma-Aldrich). SDS-PAGE was performed with 20 µg protein per sample, transferred to a polyvinylidene difluoride membrane (Immobilon), blocked with 5% BSA/TBST for 3 h at room temperature and incubated with anti-β-actin (Santa Cruz Biotechnology, sc-47778; 1 to 500), anti-cortactin (Abcam, ab33333; 1 to 1000), anti-E-cadherin (BD Transduction Laboratories, 610181; 1 to 1000), anti-FAK (Santa Cruz Biotechnology, sc-271126, 1 to 100), anti-phospho-FAK (Tyr397) (Invitrogen, 44625 G, 1 to 100) anti-N-cadherin (BD Transduction Laboratories, 610920; 1 to 500) and anti-Tks5 (Millipore, MABT336; 1 to 500) antibodies diluted in 5% BSA/TBST overnight at 4 °C. The membranes were then incubated with HRP conjugated anti-mouse or anti-rabbit IgG (Cell Signaling Technologies; 1 to 5000) antibodies diluted in 5% non-fat milk/TBST for 1 h at room temperature and proteins were visualized using chemiluminescence detection reagents (WesternBright, Advansta) and blot scanner (C-DiGit, LI-COR).

**Tissue analyses**. To form tumors in mice, 200,000 cells were suspended in 100 µl of 20% collagen I in PBS and injected orthotopically into the mammary fat pad of 7-week-old female mice. For inoculation with mixtures of 4T1 or MDA-MB-231 cells, the cell ratio was 1:1, and 1:300 for 4T1:67NR. When the tumor diameter reached 8–12 mm, 14–20 days in Balb/cJ mice for 4T1 and 67NR, 8-10 weeks in SCID mice for MDA-MB-231, the animals were sacrificed and the tumors and lungs harvested. For lung clonogenic assay (https://doi.org/10.5281/zenodo.6639302), lungs were minced, digested in a collagenase type IV/elastase cocktail (Worthington Biochemical) and filtered through a 70 µm mesh. Then, the cell suspension from each lung was split into two tissue culture plates. One plate was incubated with a combination of 6-thioguanine (Cayman Chemicals) and puromycin, for growth of Tks5-CTL and Tks5-KD cells. The other plate was incubated with a combination of 6-thioguanine, puromycin and geneticin for growth of 4T1 Tks5-KD cells only, or with a combination of 0.5 µg/ml puromycin and 500 µg/ml geneticin (Invitrogen) for growth of MDA-MB-231 D2-KD. After 14 days at 37 °C and 5% CO2, the colonies were fixed in methanol, stained using 0.03% (w/v) methylene blue (Sigma-Aldrich) and counted (4T1 and 67NR), or counted using fluorescent labeling (D2-KD).

To perform Western blots on tumors, Tks5-KD tumors were harvested, minced and digested in a freshly prepared collagenase type III (Worthington Biochemical) solution in HBSS, filtered through a 70 μm mesh and cultured for 7 days in the medium supplemented with 6-thioguanine and geneticin. Cells were lysed as described above.

To image tumor sections, tumors were fixed in 4% PFA, 4 °C overnight, washed with ice cold PBS for 1 h, transferred to 30% sucrose solution and incubated overnight at 4 °C, embedded, frozen in O.C.T. and cut at 6 μm thickness. To increase 67NR-GFP signal, mixed 4T1-67NR tumors were labeled with anti-GFP antibody primary antibody (ab13970, 1 to 100) and goat anti-Chicken IgY H&L (Alexa Fluor® 488, ab150169), and nuclei were stained with DAPI.

**Statistics and reproducibility**. RStudio software was used to perform all statistical analyses. The distribution of each data set was analyzed, and the Shapiro-Wilk test was performed to test for normality. For normally distributed data sets, an $F$ test was performed to compare the variances of two data sets. Based on the results from the $F$ test, a Welch two-sample $t$-test or a two-sample $t$-test was done to compare the means of the two data sets. For non-normally distributed data sets a Wilcoxon rank sum test was performed to compare the two data sets. Unless stated otherwise, all tests were performed using unpaired and two-sided criteria. All data are represented as line graphs or boxplots with median (line), 25th/75th percentiles (boxes) and maximum/minimum (whiskers). Statistical significance was defined as $*p < 0.05$, $**p < 0.01$, and $***p < 0.001$. Additional information on the metrics and statistics can be found in the source data file. All data are available upon request.

**Reporting summary**. Further information on research design is available in the Nature Research Reporting Summary linked to this article.

## Data availability

All data generated or analyzed during this study are included in this published article and its supplementary information as Supplementary Data 1. For legends of Source Data and Movies, please see Description of Additional Supplementary Materials. Plasmid pmScarlet-H-C1 was a gift from Dorus Gadella (Addgene plasmid # 85043), while pEGFP-puro was a gift from Michael McVoy (Addgene plasmid # 45561). Uncropped western blots are available in Supplementary Figs. S13–S27.

## Code availability

All code for data processing and analysis associated with the current submission is available at https://github.com/tuzellab/cooperative_invasion and (https://doi.org/10.5281/zenodo.6639302).

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

## Acknowledgements

We would like to thank the flow cytometry core at Lewis Katz School of Medicine for assistance with cell sorting, and the members of Temple Bioengineering and Fox Chase Cancer Center Biology for valuable discussions. Funding was provided by NIH R00 CA172360, R01 CA230777 (B.G.) and R01 GM121679 (E.T.); American Cancer Society Research Scholar Grant 134415-RSG-20-034-01-CSM (B.G.) and Conquer Cancer Now/ Young Investigator Award (B.G.).

## Author contributions

Conceptualization: L.P., B.G., Data acquisition: L.P., E.B., B.B., B.G., Analysis: L.P., E.B., B.G., E.T., Supervision: B.G., Writing: L.P., E.B., B.B., E.T., B.G.

## Competing interests

The authors declare no competing interests.
