## [Peer Review File · Communications Biology]

Reviewers' comments:

Reviewer #1 (Remarks to the Author):

The manuscript by Perrin and co-authors describes how cancer clones with different invasive abilities can cooperate during invasion. By using cancer spheroid model where 4T1 (invasive) and 67NR (no invasive) cells are mixed together, they found that differences in persistence motility and cell-ECM sensitivity drive the cell sorting and enrichment of invasive 4T1 cells at the spheroid periphery. Moreover, this study shows that lesser invasive follower cells don't require E-cadherin mediated cell-cell attachment to follow the leading edge more invasive cells. Instead, following cell sorting, 4T1 cells form invadopodia in order to create microtracks in the collagen matrix and lead 67NR cells in cooperative invasion. In the last part of the paper, the authors study the role of cooperative invasion on breast cancer metastasis.

In general, this is a very interesting study with state-of-art imaging technology and innovative cell mixing experiments. The combination of experiments with 3D culture and in vivo tumor xenografts are comprehensive. The data are of high quality, well controlled and supportive of the main conclusion. The following suggestions should be considered while revising the manuscript.

Major revisions:

1. In Fig 1E the authors show that the difference in the invasive ability of 4T1 and 67NR cells is caused by a disparity in their invadopodia function. It would be nice to add F-actin staining in fig 1E. This can help to clarify whether 67NR cells do not form invadopodia at all or the structure are formed but they do not degrade.
2. Fig 1H: It is difficult to visualize colocalization of TKS5/Col-3/4 and F-actin in the figure. Maybe add more arrows in the inset to highlight the spots of co-localization?
3. Figure S2 D: the figure legend states that quantification is between L-L and F-F however the graph is labelled as L-F and F-F
4. Fig S2 C and D do not support the L-F junction enrichment of E-cad in specific. Even though representative pictures do show enrichment at these junctions. The quantitative data presented show enrichment at all junctions in general. Along the same line, were similar findings observed in the mixed spheroids between 4T1 and 4T1 with Tks5 knockdown?
5. Line 250-252: the text states that: "all the 4T1 cells that crossed from the gelatin onto the poly-L-lysine area migrated back to the gelatin (Movie S10)." Since the movie shows only one cell that migrates to poly-L-Lysine and came back to gelatin, the author should provide a quantification to validate this phenomenon.
6. Line 260-261: the text states that the difference in the adhesion strength of 4T1 cells on gelatin and poly-L-Lysine is correlated with high level of FAK. It is important to probe for phospho-FAK to determine if it is correlated with FAK kinase activity, not just expression. They should clarify if the results of the western blot in figS5E are obtained from cells plated on gelatin, poly-L-Lysine coated dishes or regular plates.
7. In fig 5.D, the authors show that invadopodia are required in leader cells to promote collective invasion in 3D, confirming the data obtained in 2D in fig. 1. Although the results obtained by using the 4T1 Tks5 KD cells are really clear, it would be nice if the authors could show representative micrographs of a strand in 3D where it is possible to observe invadopodia in the 4T1 CTR leader cells.
8. In the discussion, the author concludes that the invasive cells can sort from and lead non-invasive cells during cooperative invasion and metastasis. However, the in vivo data shown in fig. 5K seem to suggest that the cell sorting is not an essential event for the metastatic process. Although the cell sorting did not occur in mixed spheroids of 4T1-KD and 4T1 or 4T1 CTR cells (as shown in Fig S9A), when injected in vivo, the 4T1-KD are able to form metastasis. In contrast to this, when the cell sorting occurs, like in the case of the mixed spheroids of 4T1 and 67NR cells, the 67NR cells are not able to form lung metastasis.

Reviewer #2 (Remarks to the Author):

In the present manuscript, Perrin et al. examined the possibility of cooperative invasion using cultured spheroids of mixed breast cancer cells with different properties. They noticed that, in those spheroids, 4T1 cells spontaneously sorted from 67NR cells to be located in the outer margin, invading into the extracellular matrix. Then they proposed that differential cell-matrix adhesion and invasion in these cells were critical for their sorting, as demonstrated by the use of different extracellular matrixes and the blockage of invadopodia with genetic and pharmacological tools. Finally, using the orthotopic injection model, they suggested that, in tumor mass composed of heterogeneous cancer cells, those with invasive properties might help those without invasion capabilities to metastasize, supporting the existence of cooperative invasion during cancer progression.

This group recruited several well-designed experiments to verify the interesting idea of cooperative invasion. The manuscript is well written with clear logical flow and data presentation. I just have some concerns about their data interpretation and conclusion, as described below in specific points:

1. Starting from Line #52 in Page 4, the authors predicted that 4T1 cells should be located inside 67NR if differential adhesion hypothesis for cell sorting was correct. Since they saw the opposite result, i.e. 4T1 was outside 67NR in the spheroids, they suggested that the main sorting power was not differential cell-cell adhesion but cell-matrix adhesion and invasion. The authors then used several experiments with perturbation of cell-matrix adhesion or cell invasion to support their points. However, in my opinion, results from those experiments, though beautiful, did not negate the involvement of differential cell-cell adhesion for cell sorting.

(1) As shown by the authors, 4T1 cells adhered and invaded the extracellular matrix better than 67NR cells. It is therefore not surprising to see 4T1 cells were located outside 67NR cells in the spheroids by their natural properties.

(2) Since 4T1 cells were not mingled with but separated from 67NR cells in the spheroids indicating the existence of cell sorting, the authors thought that cell-matrix adhesion and invasion were responsible for such sorting. However, if cell-cell adhesion were not involved, we would not see the separation of 4T1 from 67NR cells. Instead, we would see that 4T1 were located both at the center and margin of spheroids while 67NR cells existed only at the center of spheroids.

(3) Although the authors used several experiments in Fig. 3 and Fig. 4 to argue that cell-matrix adhesion and invasion were involved in cell sorting, those results just supported that 4T1 and 67NR had distinct adhesion and invasion behaviors. When 4T1 cells could not adhere to the matrix and invade, it would definitely stay with 67NR cells in their initial positions of the spheroids. Therefore sorting would definitely not happen since cells could not even move.

(4) Therefore, I do not think the authors' experimental results supported their arguments that cell-matrix adhesion and invasion were more important than cell-cell adhesion for cell sorting. Experiments to exclude the importance of cell-cell adhesion are needed if they want to keep the arguments. Otherwise the authors may want to expand their model to include the involvement of differential cell-cell adhesion together with cell-matrix adhesion & invasion.

(5) Moreover, experiments with perturbations of cell-cell adhesion will elucidate whether or not collective movement is important for cancer cell invasion from this 3D spheroid model. So it will help clarifying the existence and importance of cooperative invasion. I strongly suggest the authors to include those experiments.

2. In Fig. 4 and Fig. 5, the authors argued that in mixed spheroids or tumor masses, cells with invasive power led those without invasion capabilities to cause efficient invasion. I am a bit worried that this is an over-statement.

(1) Although, in the 3D spheroid model (Fig. 4), increased movement of 67NR was seen with the presence of 4T1 cells, equivalent phenomena were not seen in the orthotopic implantation model, i.e. 4T1 cells did not help the movement of 67NR cells to metastatic colonies (Fig. 5I).

(2) Even if non-invasive cells can be helped by invasive cells for distant metastasis, I wonder if it is important for cancer progression since the non-invasive cells may have no contribution to morbidity and mortality, both of which might be totally caused by invasive cells.

(3) Therefore, to support their statement, the authors may need experiments to show that non-invasive cells increase the metastatic power of invasive cells, and / or that the addition of non-invasive cells increase the progression of cancer and mortality of animals. Otherwise their conclusion needs to be downplayed.

Reviewer #3 (Remarks to the Author):

please see attached file

Invadopodia enable cooperative invasion and metastasis of breast cancer cells
Perrin et al.

In this manuscript the authors nicely show that cooperative invasion requires the formation of invadopodia in the leader cells, and that a small number of leader cells may be enough to facilitate collective invasion of a larger cell group, including non-invadopodia forming cancer cells. It also shows that invadopodia activity may not be necessary for sorting cells within a given spheroid. A careful examination of cell-cell interactions during invasion in a 3D collagen model is timely and significant.

Comments:

1. To further ensure robustness of data and conclusions, a second Tks5 shRNA construct or a Tks5 rescue should be used.
2. It would be important to ensure that the KD is maintained over the course of the assays, including lung colonization.
3. It would be important to have some indication of how the cell mixtures survive/proliferate in vivo in the primary tumor in order to ensure that the effects on lung colonization happen during metastasis and not earlier.

Reviewer #1

Major revisions:

1. In Fig 1E the authors show that the difference in the invasive ability of 4T1 and 67NR cells is caused by a disparity in their invadopodia function. It would be nice to add F-actin staining in fig 1E. This can help to clarify whether 67NR cells do not form invadopodia at all or the structure are formed but they do not degrade.

We have now added a Figure S3, which shows that the numbers of invadopodia precursors in 4T1 and 67NR cells, defined as colocalization of F-actin + Tks5, are similar.

Accompanying text was added to line 131:

Puncta of co-localized Tks5 and F-actin, indicative of invadopodia precursors, were present in both 4T1 and 67NR cells, at similar levels (Fig. S3A, B). Altogether, these results suggests that invadopodia precursor fail to mature in 67NR cells.

Figure S3. (A) 4T1 (top) and 67NR (bottom) cells cultured on fluorescent gelatin (not shown). Tks5 (green) and F-actin (phalloidin, magenta) were stained. Yellow arrowheads indicate invadopodia precursors. Scale bars: 20 μm . (B) Number of invadopodia precursors (Tks5 + F-actin) per cell from (A).

2. Fig 1H: It is difficult to visualize colocalization of TKS5/Col-3/4 and F-actin in the figure. Maybe add more arrows in the inset to highlight the spots of co-localization?

We have now added arrows in the inset, pointing out the colocalization of markers in the leader cell of the invasive strand.

3. Figure S2 D: the figure legend states that quantification is between L-L and F-F however the graph is labelled as L-F and F-F

We thank the reviewer for noticing this error, we have now corrected the legend.

4. Fig S2 C and D do not support the L-F junction enrichment of E-cad in specific. Even though representative pictures do show enrichment at these junctions. The quantitative data presented show enrichment at all junctions in general. Along the same line, were similar findings observed in the mixed spheroids between 4T1 and 4T1 with Tks5 knockdown?

We thank the reviewer for pointing this inconsistency. We rephrased the text in line 112, to:

Further, in the 4T1 spheroids, we found E-cadherin to be enriched at all cell-cell junctions, namely between leader and follower cells as well as between follower and follower cells. This verified that the integrity of E-cadherin-mediated cell-cell junctions was maintained during invasion (Fig. S2C, D).

We have also added a supplementary Figure S9, showing similar findings were observed at cell-cell junctions between Tks5-CTL and Tks5-KD cells.

5. Line 250-252: the text states that: “all the 4T1 cells that crossed from the gelatin onto the poly-L-lysine area migrated back to the gelatin (Movie S10).” Since the movie show only one cell that migrates to poly-L-Lysine and came back to gelatin, the author should provide a quantification to validate this phenomenon.

We thank the reviewer for this comment, we rephrased the text, now in line 258:

...in the rare event of 4T1 cells crossing from the gelatin onto the poly-L-lysine area, cell ended up migrating back to the gelatin.

6. Line 260-261: the text states that the difference in the adhesion strength of 4T1 cells on gelatin and poly-L-Lysine is correlated with high level of FAK. It is important to probe for phospho-FAK to determine it is correlated with FAK kinase activity, not just expression.

We thank the reviewer for this suggestion and have now included phospho-FAK in Figure S6E, showing 4T1 have higher p-FAK compared to 67NR cells.

Accompanying text was added on line 267:

This affinity to ECM is in line with the higher levels of both FAK and p-FAK observed in 4T1 cells than in 67NR cells (Fig. S6E).

They should clarify if the results of the western blot in figS5E are obtained from cells plated on gelatin, poly-L-Lysine coated dishes or regular plates.

For western blot experiments, cells were plated onto poly-L-lysine coated dishes. We now added this information into the first sentence of Western blot assay section in Materials and Methods, at line 718:

Cells were plated onto poly-L-lysine coated dishes and cultured to 80% confluency.

7. In fig 5.D, the authors show that invadopodia are required in leader cells to promote collective invasion in 3D, confirming the data obtained in 2D in fig. 1. Although the results obtained by using the 4T1 Tks5 KD cells are really clear, it would be nice if the authors could show representative micrographs of a strand in 3D where it is possible to observe invadopodia in the 4T1 CTL leader cells.

We have now added the following text on line 287:

Importantly, in mixed spheroids, the majority of strands was led by 4T1 cells (Fig. 5D), which assemble invadopodia (Fig. 1H).

8. In the discussion, the author conclude that the invasive cells can sort from and lead non-invasive cells during cooperative invasion and metastasis. However, the in vivo data showed in fig. 5K seem to suggest that the cell sorting is not an essential event for the metastatic process.

Although the cell sorting did not occur in mixed spheroids of 4T1-KD and 4T1 or 4T1 CTR cells (as shown in Fig S9A), when injected in vivo, the 4T1-KD are able to form metastasis. In contrast to this, when the cell sorting occurs, like in the case of the mixed spheroids of 4T1 and 6TNR cells, the 6TNR cells are not able to form lung metastasis.

We have now modified the text in the Discussion, line 427 to say:

We find that in the spheroids where highly-persistent, invasive cells sensitive to ECM are mixed with non-invasive, randomly moving cells, sorting and accumulation of invasive cells at the spheroid-ECM interface precedes cooperative invasion.

And line 434 to say:

Importantly, while cell sorting may be catalytic, it does not seem to be essential for the cooperative metastasis in our system. While mixed 4T1 Tks5-KD and Tks5-CTL cells engage in cooperative metastasis, but do not sort, mixes of 4T1 and 67NR cells sort but do not metastasize cooperatively.

Reviewer #2 (Remarks to the Author):

1. Starting from Line #52 in Page 4, the authors predicted that 4T1 cells should be located inside 67NR if differential adhesion hypothesis for cell sorting was correct. Since they saw the opposite result, i.e. 4T1 was outside 67NR in the spheroids, they suggested that the main sorting power was not differential cell-cell adhesion but cell-matrix adhesion and invasion. The authors then used several experiments with perturbation of cell-matrix adhesion or cell invasion to support their points. However, in my opinion, results from those experiments, though beautiful, did not negate the involvement of differential cell-cell adhesion for cell sorting.

(1) As shown by the authors, 4T1 cells adhered and invaded the extracellular matrix better than 67NR cells. It is therefore not surprising to see 4T1 cells were located outside 67NR cells in the spheroids by their natural properties.

(2) Since 4T1 cells were not mingled with but separated from 67NR cells in the spheroids indicating the existence of cell sorting, the authors thought that cell-matrix adhesion and invasion were responsible for such sorting. However, if cell-cell adhesion were not involved, we would not see the separation of 4T1 from 67NR cells.

Instead, we would see that 4T1 were located both at the center and margin of spheroids while 67NR cells existed only at the center of spheroids.

(3) Although the authors used several experiments in Fig. 3 and Fig. 4 to argue that cell-matrix adhesion and invasion were involved in cell sorting, those results just supported that 4T1 and 67NR had distinct adhesion and invasion behaviors. When 4T1 cells could not adhere to the matrix and invade, it would definitely stay with 67NR cells in their initial positions of the spheroids. Therefore sorting would definitely not happen since cells could not even move.

(4) Therefore, I do not think the authors' experimental results supported their arguments that cell-matrix adhesion and invasion were more important than cell-cell adhesion for cell sorting. Experiments to exclude the importance of cell-cell adhesion are needed if they want to keep the arguments. Otherwise the authors may want to expand their model to include the involvement of differential cell-cell adhesion together with cell-matrix adhesion & invasion.

(5) Moreover, experiments with perturbations of cell-cell adhesion will elucidate whether or not collective movement is important for cancer cell invasion from this 3D spheroid model. So it will help clarifying the existence and importance of cooperative invasion. I strongly suggest the authors to include those experiments.

We have now included experiments which use a) E-cadherin blocking antibody or b) stable E-cadherin KD cell lines, with two different shRNA constructs Ecad-KD1 and Ecad-KD2 into Figure 3A-H, and Figure 5H-K. In summary, loss of E-cadherin did not affect the cell sorting (Figure 3A, B and 3G, H), confirming our previous conclusions. In addition, the 3D spheroid invasion of 4T1 cells was shifted from collective to single-cell invasion mode (Figure 3F). Interestingly, the non-invasive 67NR were unable to cooperatively invade by following the single 4T1 cells (Figure 5K).

To describe these results, we added the following paragraphs in the Results section, at line 210:

Cell sorting is E-cadherin independent

While self-organization and patterning of cells is heavily studied in embryonic tissues and during development, so far, only a few studies tested mixtures of two or more cell types in the cancer spheroid model (19, 20). Most studies reported that cells sort based on the differential adhesion hypothesis, which predicts sorting based on differences in the intercellular adhesiveness, with cells that exhibit the strongest cell-cell adhesions positioned in the center (35). Based on the

expression of cadherins for the 4T1/67NR pair, the differential adhesion hypothesis predicts that in a 3D spheroid, E-cadherin-expressing 4T1 will sort from N-cadherin-expressing 67NR cells, with 4T1 cells located in the center and 67NR cells surrounding them. In contrast, our data demonstrates the opposite pattern (Fig. 2A, B), suggesting that the differential adhesion hypothesis may not apply in our model. To confirm this, we inhibited cell-cell junctions in 4T1 cells via E-cadherin blocking antibody (Fig. 3A). Interestingly, blocking E-cadherin did not eliminate cell sorting between 4T1 and 67NR cells, but delayed it by 1 day (Fig. 3B). To confirm this result, we generated stable E-cadherin knock down cell lines (Ecad-KD1 and Ecad-KD2; Fig. 3C-E). According to Western blot analysis decrease in E-cadherin expression was 36.3% for Ecad-KD1 and 96.8% for Ecad-KD2. Both Ecad-KD cell lines sorted out from the 67NR-GFP cells, and accumulated to the spheroid edge by day 3 post-embedding (Fig. 3G, H). Overall, this suggests that, in our model, cell sorting is independent from E-cadherin.

Fig. 3 Disruption of cell-cell adhesions does not affect cell sorting. Mixed spheroids, at a 1:50 ratio, imaged on day 4 post-embedding. Spheroids were treated with (+Ab, right panel) or without (-Ab, left panel) an E-cadherin blocking antibody. Scale bar: 100 μ m. **(B)** DI for 4T1-mScar (magenta boxes) and 67NR-GFP (green boxes) cells from spheroids in (A). $P=2.78\times 10^{-5}$, $<2.2\times 10^{-16}$ and 1.12×10^{-12} respectively, by the Wilcoxon rank sum test. **(C)** Ecad-CTL, Ecad-KD1 and -KD2 cells in 2D. E-cadherin (green, bottom panels) and nuclei (gray, top panels) are shown. **(D)** Integrated density of the E-cadherin signal from cells in (C). $P=4.85\times 10^{-12}$ and $<2.2\times 10^{-16}$, by the Wilcoxon rank sum test. **(E)** Western-blot of E-cadherin expression in Ecad-CTL, -KD1 and -KD2 cells. β -actin is used as a loading control. **(F)** Number of strands per spheroid (black) and number of single cells per spheroid (blue) for Ecad-CTL, -KD1 or -KD2 spheroids. The red empty symbols indicate zero values. $P=9.01\times 10^{-10}$, 1.06×10^{-3} and 1.74×10^{-5} , by the t-test. **(G)** Mixed spheroids, at a 1:50 ratio, imaged on day 4 post-embedding. Ecad-CTL, -KD1 or -KD2 cells were used. Scale bar: 100 μ m. **(H)** DI for Ecad-CTL, -KD1 and -KD2 (magenta boxes) and 67NR-GFP (green boxes) cells from spheroids in (G). $P=4.72\times 10^{-5}$, $<2.20\times 10^{-16}$, 2.21×10^{-5} , 2.12×10^{-15} , 2.05×10^{-13} and $<2.20\times 10^{-16}$, by the Wilcoxon rank sum test.

And line 316:

We wondered if the inhibition of E-cadherin, which had no effect on cell sorting (Fig. 3F, G), can affect cooperative invasion. To test this, we imaged 4T1, Ecad-CTL, -KD1 and -KD2 spheroids on day 6, when invasion occurs (Fig. 5H). While the Ecad-KD1 cell line showed a partial transition from collective to single-cell invasion, with invasive strands as well as single invasive cells, a complete transition to single cell invasion was observed for Ecad-KD2. Similarly, in the mixed spheroids containing 67NR cells with either Ecad-CTL or -KDs, while Ecad-CTL and Ecad-KD1 cells formed invasive strands, both Ecad-KD cell lines exhibited single cell invasion (Fig. 5I-K). To quantify occurrences of cooperative invasion in both collective and single cell modes of invasion, we measured the number of 67NR followers present either in strands (Fig. 5J), or as single cells (Fig. 5K). Our results demonstrate that while 67NR can follow Ecad-KD1 cells, which maintain invasive strands (Fig. 5I), in the presence of complete transition to single cell invasion, such as in mixed spheroids of Ecad-KD2 and 67NR, cooperative invasion is lost (Fig. 5K).

2. In Fig. 4 and Fig. 5, the authors argued that in mixed spheroids or tumor masses, cells with invasive power led those without invasion capabilities to cause efficient invasion. I am a bit worried that this is an over-statement. (1) Although, in the 3D spheroid model (Fig. 4), increased movement of 67NR was seen with the presence of 4T1 cells, equivalent phenomena were not seen in the orthotopic implantation model, i.e. 4T1 cells did not help the movement of 67NR cells to metastatic colonies (Fig. 5I). (2) Even if non-invasive cells can be helped by invasive cells for distant metastasis, I wonder if it is important for cancer progression since the non-invasive cells may have no contribution to morbidity and mortality, both of which might be totally caused by invasive cells. (3) Therefore, to support their statement, the authors may need experiments to show that non-invasive cells increase the metastatic power of invasive cells, and / or that the addition of non-invasive cells increase the progression of cancer and mortality of animals. Otherwise their conclusion needs to be downplayed.

Once cells reach the lung and start growing into a metastatic colony, cell proliferation locally obstructs or blocks the lung airways, lowering the patient life expectancy. As Tks5-KD and Tks5-

CTL proliferate at similar rates (Gligorijevic *et al*, Plos Biol 2014), we expect that non-invasive and invasive lung colonies contribute to morbidity at similar levels.

We have now included additional data on MDA-MB-231 in Supplementary Figure 10I, and a paragraph in the Discussion (line 452), clarifying our results:

Our findings on cooperative metastasis of 4T1 Tks5-KD with Tks5-CTL are in alignment with the previous demonstration of dissemination of epithelial cell clusters (3). In contrast, 67NR cells do not cooperatively metastasize when mixed with 4T1. In addition, MDA-MB-231 Tks5-KD (D2-KD) cells do not cooperatively metastasize with MDA-MB-231 cells. Both of these cell lines lack E-cadherin, suggesting that the cooperative metastasis is dependent on strong E-cadherin-based cell-cell interactions. Interestingly, cooperative invasion in 3D spheroid assay does not pose such a requirement, as both 67NR and D2-KD cells successfully follow their invasive counterparts. The likely reason is that the collagen deformations generated by the invasive leader cells are plastic (permanent), allowing either cell strands or individual cells to migrate through them (39). In contrast, deformations that leader cells generate to the blood vessel wall are elastic (transient), and followers can cross blood vessel walls only if they exhibit strong cell-cell adhesions to the leader.

Reviewer #3 (Remarks to the Author):

1. To further ensure robustness of data and conclusions, a second Tks5 shRNA construct or a Tks5 rescue should be used.
2. It would be important to ensure that the KD is maintained over the course of the assays, including lung colonization.

We thank the reviewer for this suggestion. Data from the second Tks5 shRNA, -KD2 construct, confirming the initial KD results, is now in the new Figure S11.

Figure S11. (A) Tks5 expression in Tks5-CTL and Tks5-KD2 cells. Knockdown efficiency is 81.6% (B) Degradation area (μm^2) per cell for Tks5-CTL and Tks5-KD2 cells plated on gelatin. $P=7.12 \times 10^{-7}$, by the Wilcoxon rank sum test. (C) Day 2 images of spheroids made with Tks5-CTL, -KD2 or a mixture (1:1 ratio) of Tks5-CTL and -KD2 cells. Scale bar: 100 μm . (D-F)

Number of strands per spheroid (D), number of strands containing Tks5-KD2 cells (E), and the percentage of strands led by Tks5-CTL and -KD2 cells (F) in spheroids from (C). $P=4.26 \times 10^{-6}$ and $P=4.78 \times 10^{-5}$, by the Wilcoxon rank sum test in (E). $P=0.0404$, by the t-test in (F). (G) Day 6 images of mixed spheroids made with 67NR-GFP and 4T1-mScarlet or Tks5-KD2 cells, at a 1:50 ratio. Scale bars: 100 μm . (H) Number of strands per spheroid from (G). $P=1.40 \times 10^{-4}$, by the Wilcoxon rank sum test. (I) Tks5 expression in 4T1-mScarlet (4T1-mScar) or Tks5-KD2 tumors. GAPDH was used as a loading control. (J) Number of lung colonies per cm^2 for mice inoculated with Tks5-CTL, Tks5-KD2 or a mixture of Tks5-CTL and Tks5-KD2 cells. $P=0.044$, by the t-test.

Accompanying text was added on line 353:

Finally, we validated the conclusion with a different Tks5 shRNA sequence, Tks5-KD2, which provided results similar to Tks5-KD (Fig. S11).

In the Figure S11I, we also show that KD is maintained in the primary tumors, by Western Blot of the dissociated Tks5-KD tumors. To ensure that the KD is maintained in the lung colonies, we have grown the Tks5-KD colonies under continuous geneticin and puromycin selection, which eliminated all the Tks5-expressing cells. This was now better explained in Materials and Methods, section **Tissue Analyses**, line 733:

...the cell suspension from each lung was split into two tissue culture plates. One plate was incubated with a combination of 6-thioguanine (Cayman Chemicals) and puromycin, for growth of Tks5-CTL and Tks5-KD cells. The other plate was incubated with a combination of 6-thioguanine, puromycin and geneticin for growth of 4T1 Tks5-KD cells only, or with a combination of 0.5 $\mu\text{g}/\text{ml}$ puromycin and 500 $\mu\text{g}/\text{ml}$ geneticin (Invitrogen) for growth of MDA-MB-231 D2-KD.

3. It would be important to have some indication of how the cell mixtures survive/proliferate in vivo in the primary tumor in order to ensure that the effects on lung colonization happen during metastasis and not earlier.

We thank the reviewer for this suggestion. Images of the mixed tumor containing 67NR-GFP and 4T1-mScar are now included in the new Figure 6J.

In addition, images of the mixed tumor containing mScarlet-MDA-MB-231 (Scar-CTL) and Dendra2-MDA-MB-231-Tks5KD (D2-KD) are now included in the new Figure S10H.

Accompanying text was added on line 377:

To determine which cell type was growing metastatic lung colonies, we leveraged the differences in fluorescent protein expression or drug sensitivity. We confirmed that the Tks5 knock down was maintained in the Tks5-KD tumors (Fig. S11I). We found that the Tks5-KD cell lines, in 4T1 or MDA-MB-231 background, and 67NR cells were not capable of lung metastasis, in contrast to control and wild type cells (Fig. 6K, Fig. S10I, Fig. S11J). When 4T1 and 67NR cells were co-injected, 67NR did not metastasize (Fig. 6L). This was not due to 4T1 cells taking over as 67NR cells were present in the tumor tissue (Fig. 6J). Similarly, the co-injection of MDA-MB-231 control and its corresponding Tks5 knockdown, D2-KD, demonstrated that only control cells were capable of metastasis (Fig. S10H, I).

REVIEWERS' COMMENTS:

Reviewer #1 (Remarks to the Author):

The revision has successfully addressed the issues raised in previous review. The data presented in the revision support the conclusions stated in the manuscript very well.

Reviewer #2 (Remarks to the Author):

The authors have clarified my concerns. I have no more questions.

Reviewer #3 (Remarks to the Author):

The authors have addressed all the concerns raised in the review process. I recommend this article for publication.